# Integrated systems approach defines the antiviral pathways conferring protection by the RV144 HIV vaccine

Slim Fourati[1], Susan Pereira Ribeiro[1], Filipa Blasco Tavares Pereira Lopes [1], Aarthi Talla[1], Francois Lefebvre[2], Mark Cameron [3], J. Kaewkungwal[4], P. Pitisuttithum[4], S. Nitayaphan[5], S. Rerks-Ngarm[6], Jerome H. Kim[7,8], Rasmi Thomas[7], Peter B. Gilbert[9], Georgia D. Tomaras[10], Richard A. Koup[11], Nelson L. Michael[7], M. Juliana McElrath[9], Raphael Gottardo[9] & Rafick-Pierre Sékaly[1]

The RV144 vaccine trial showed reduced risk of HIV-1 acquisition by 31.2%, although mechanisms that led to protection remain poorly understood. Here we identify transcriptional correlates for reduced HIV-1 acquisition after vaccination. We assess the transcriptomic profile of blood collected from 223 participants and 40 placebo recipients. Pathway-level analysis of HIV-1 negative vaccinees reveals that type I interferons that activate the IRF7 antiviral program and type II interferon-stimulated genes implicated in antigen-presentation are both associated with a reduced risk of HIV-1 acquisition. In contrast, genes upstream and downstream of NF-κB, mTORC1 and host genes required for viral infection are associated with an increased risk of HIV-1 acquisition among vaccinees and placebo recipients, defining a vaccine independent association with HIV-1 acquisition. Our transcriptomic analysis of RV144 trial samples identifies IRF7 as a mediator of protection and the activation of mTORC1 as a correlate of the risk of HIV-1 acquisition.

[1] Department of Pathology, Case Western Reserve University, Cleveland, OH 44106, USA. [2] Canadian Center for Computational Genomics, Montréal, QC H3A 0G1Canada. [3] Department of Epidemiology and Biostatistics, Case Western Reserve University, Cleveland, OH 44106, USA. [4] Faculty of Tropical Medicine, Mahidol University, Bangkok 10400, Thailand. [5] Royal Thai Army, Armed Forces Research Institute of Medical Sciences, Bangkok 10400, Thailand. [6] Department of Disease Control, Ministry of Public Health, Nonthaburi 11000, Thailand. [7] Military HIV Research Program, Walter Reed Army Institute of Research, Silver Spring, MD 20910, USA. [8] International Vaccine Institute, Seoul 08826, Korea. [9] Vaccine and Infectious Disease Division, Fred Hutchinson Cancer Research Center, Seattle, WA 98109, USA. [10] Duke Human Vaccine Institute, Duke University, Durham, NC 27710, USA. [11] Vaccine Research Center, US National Institutes of Health, Bethesda, MD 20892, USA. Correspondence and requests for materials should be addressed to R.-P.S. (email: rafick.sekaly@case.edu)

The RV144 trial evaluated the efficacy of ALVAC-HIV (vCP1521) prime and AIDSVAX B/E (gp120) boost strategy adjuvanted in alum to prevent human immunodeficiency virus 1 (HIV-1) acquisition. Participants enrolled in the RV144 clinical trial were followed up to 3 years after a series of four immunizations. The vaccine reduced the risk of HIV-1 acquisition at 3 years following completion of the vaccination series by 31.2% when compared to placebo (modified intention-to-treat analysis, Likelihood-ratio test: $p = 0.0385$)[1]. Correlates of risk studies showed that two nonneutralizing antibody responses measured 2 weeks after vaccination were associated with HIV-1 acquisition: levels of IgA recognizing the Envelope (Env) region of HIV-1 associated with a higher risk of HIV-1 acquisition (similar to the risk of placebo recipients) and levels of IgG recognizing the V1/V2 regions of HIV-1 Env associated with a decreased risk of HIV-1 acquisition[2]. More recently, CD4 + T cells polyfunctionality measured in response to Env stimulation (i.e., polyfunctionality score (PFS)) was associated with a decreased risk of HIV-1 acquisition in RV144 vaccinees[3], while two host human leukocyte antigen (HLA) alleles (DBQ1*06 and DPB1*13) were shown to modulate the HIV-specific antibody responses associated with HIV-1 acquisition[4]. The benefit of combining these correlates, that underlie different arms of the immune response (humoral and cellular), to predict the risk of HIV-1 acquisition among RV144 vaccinees has not been assessed. Moreover, a specific innate immune response that can help prime cellular and humoral immune effector mechanisms are yet to be defined.

Despite the identification of correlates of risk for the RV144 vaccine that could potentially be correlates of protection[5], the mechanisms that lead to vaccine conferred-protection are still unknown. IgG antibodies elicited by the vaccine have been suggested to trigger antibody-dependent cell-mediated cytotoxicity (ADCC) by binding Fc receptors on the surface of natural killer cells, whereas IgA antibodies compete with IgG antibodies for binding to HIV-1 Env and thus abrogate ADCC in vaccinees[6]. Conversely, the ALVAC vector has been reported to trigger cytosolic pattern-recognition receptors sensing double-stranded DNA leading to the activation of IRF3/IRF7 and the induction of an innate immune antiviral response that could prevent HIV-1 infection[7]. Understanding the mechanisms that led to RV144 vaccine-conferred protection could offer new insights into the development of more effective HIV vaccines.

In this study, we assessed the transcriptomic profile of in vitro stimulated peripheral blood mononuclear cells (PBMCs) taken from 223 vaccinees and 40 placebo-recipient two weeks after the last immunization with the RV144 vaccine (or placebo). We identified that IFNγ stimulated genes are associated with reduced risk of HIV-1 acquisition among vaccinees. Genes downstream of NF-κB and mTORC1 required for viral infection or replication are instead associated with increased risk of HIV-1 acquisition in both vaccinees and placebo recipients, with a mechanism independent of the vaccine-induced immune response.

## Results

**The RV144 vaccine induces IFNγ, NF-κB, and mTORC1 pathways**. As an initial step, a pilot study was conducted to identify (through differential gene-expression analysis) and down-select (through clustering) RV144 vaccine-induced transcriptomic signatures that would then be tested for their association with HIV-1 acquisition[2] (Fig. 1). We compared the transcriptomic profile of in vitro HIV-1 Env-stimulated PBMCs obtained pre-immunization and 2 weeks after the last immunization from 40 vaccine recipients and 10 placebo recipients; all 50

participants were HIV-1 negative at the last follow-up visit (Supplementary Tables 1 and 2)[2]. Linear regression models followed by gene set enrichment analysis (GSEA) were used to identify pathways that were differentially regulated when comparing the vaccine and placebo groups (see Methods and Supplementary Data 1–3). Linear regression revealed that the expression of 2946 genes was altered postimmunization (differentially expressed between the vaccine and placebo groups, LIMMA: moderated $t$ test $p \leq 0.05$); none of these genes showed differential expression after correction for multiple testing. GSEA identified 11 pathways significantly enriched among genes induced postimmunization compared to pre-immunization in Env-stimulated PBMCs of vaccinees while none could be detected in Env-stimulated PBMCs of placebo recipients (DB: Hallmark, GSEA: false-discovery rate (FDR) ≤ 5%, Fig. 2a). Sample level enrichment analysis (SLEA), a method allowing to average expression of genes of a given pathway per sample, was used to quantify the enrichment of those 11 pathways for each sample of the pilot study. These 11 pathways were separated into four groups based on their correlation across samples (Fig. 2b and Supplementary Fig. 1). In vaccinees, Env stimulation led to the induction of genes part of the IFNγ response pathway (englobing as well IFNα response genes), genes implicated in NF-κB signaling, genes downstream of mTORC1 as well as genes associated to allograft rejection, i.e., genes triggered by T-cell activation. Known IFNγ response genes specifically induced in RV144 vaccinees included genes implicated in antigen presentation by the major histocompatibility complex (MHC) class I (FCGR1A, HLA-B, HLA-G, ICAM1[8], TAP1) and by MHC class II (CD74, HLA-DQA1, HLA-DMA, HLA-DRB1; Supplementary Data 4). Several of those mediate antiviral responses (IFIH1[9], MYD88[10], TRAFD1[11]) or have described antiviral effects against HIV-1 (APOL6, LGALS3BP, RSAD2; Supplementary Data 4). Stimulation of PBMCs with HIV-1 Env was not associated with increased expression of these genes in placebo recipients (Fig. 2c). Likewise, several genes coding for members of the NF-κB transcriptional complex (NFKBIA, REL), genes encoding upstream activators of the NF-κB transcriptional activity (BIRC3, IL1B, RIPK2, TANK, TNF) as well as several NF-κB transcriptional targets (CXCL2, ICAM1, IL6, IL8, SDC4) were specifically induced in Env stimulated cells from RV144 vaccinees (Supplementary Data 4). Several mTORC1 downstream targets (GCLC, SCD, TFRC) were induced as well by Env stimulation of PBMCs from RV144 vaccinees (Supplementary Data 4). Together these results highlight IFNγ-, NF-κB-, and mTORC1-regulated genes as major transcriptional targets of the RV144 vaccine in Env-stimulated PBMCs.

**IFNγ pathway is associated with a reduced risk of acquisition**. To determine if changes in gene expression were associated with HIV-1 acquisition, we analyzed the transcriptomic profiles of HIV-1 Env in vitro stimulated PBMCs obtained 2 weeks after the last immunization from 183 vaccine recipients part of the case–control study[2] that included 30 participants that acquired HIV-1 after vaccination (cases) and 153 participants that remained HIV-1 negative during the 3 year follow-up (controls). In addition to case–control samples, this dataset also included 30 placebo recipients, 17 of which acquired HIV-1 after vaccination, and 13 that remained HIV-1 negative (Supplementary Tables 3 and 4). We used linear regression models and GSEA to assess the association of the four pathways induced by the RV144 vaccine with HIV-1 acquisition; this analysis was performed separately for vaccinees and placebo recipients. A total of 2058 genes were differentially expressed between participants that acquired HIV-1 and those that remained HIV-1 negative within the placebo

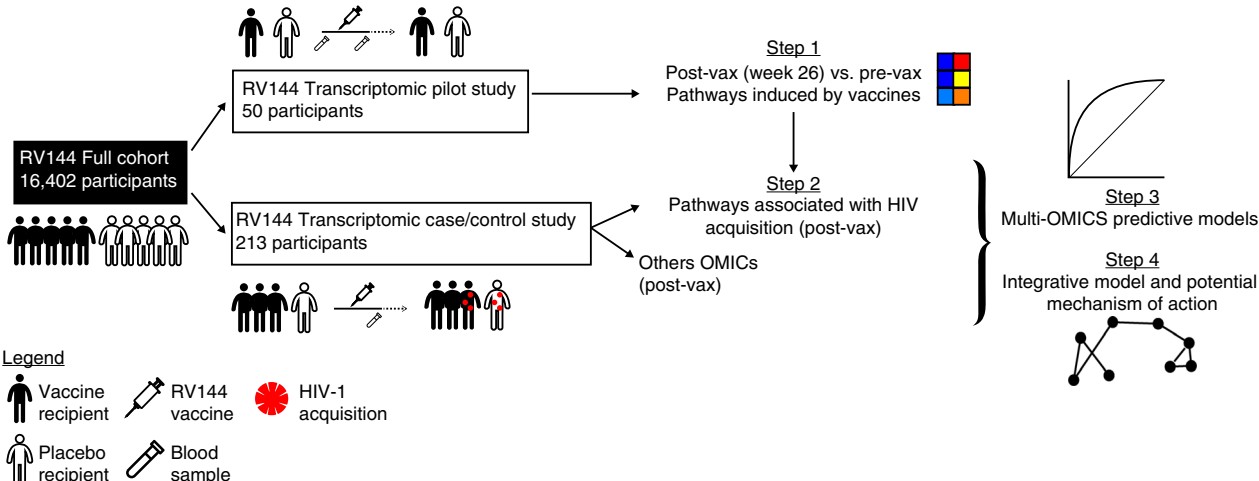

**Fig. 1** Study overview. Four analysis steps were used to identify transcriptomic markers of risk of HIV-1 acquisition among RV144 vaccinees. A first transcriptomic dataset of blood collected from 40 HIV-1 negative vaccinees and 10 HIV-1 negative placebo recipients prevaccination and 2 weeks after vaccination was used to identify pathways modulated by the RV144 vaccine (step 1). A second independent transcriptomic dataset of blood collected from 183 case–control vaccinees (including 31 infected participants) and 30 placebos (including 17 infected participants), 2 weeks after vaccination was used to identify pathways associated with HIV-1 acquisition. Logistic regression was used to build a multi-OMICS classifier of HIV-1 acquisition among RV144 vaccinees (step 3) and a projection-based integrative analysis was used to associate the different OMICS to identify mechanistic mediators of vaccine response (step 4). Three elements ("syringe [https://www.svgrepo.com/svg/10894/injecting-syringe]", "blood tube [https://www.svgrepo.com/svg/33576/blood-test]" and "man [https://www.svgrepo.com/svg/3680/standing-frontal-man-silhouette]") were modified and used in the figure under "CC-BY 4.0 [https://creativecommons.org/licenses/by/4.0/]" license

group, while 3009 genes were differentially expressed between those two groups of participants in the vaccine group (LIMMA: $t$ test $p \leq 0.05$); none of these genes remained significative after correction for multiple testing. Our analysis revealed that three out of the four pathways induced by the RV144 vaccine (described above) were significantly associated with HIV-1 acquisition in the vaccine group, namely genes part of the IFNγ response pathway, genes implicated in NF-κB signaling and genes downstream of mTORC1 (GSEA: FDR $\leq 0.05$; Fig. 3). The IFNγ response pathway was unique in its association with a lower risk of HIV-1 acquisition. In contrast, the remaining two pathways, NF-κB and mTORC1 signaling, were associated with higher risk of HIV-1 acquisition for RV144 vaccinees (Fig. 3, Supplementary Table 5, Supplementary Data 5 and 6).

The association between the induction of the IFNγ response pathway and lower risk of HIV-1 acquisition was not observed for the placebo group suggesting that the IFNγ response pathway is associated with a vaccine-conferred reduced risk of HIV-1 acquisition (Fig. 3). These IFNγ response genes included genes involved in the maturation of the MHC class II complex (*AP2A1* coding for the AP2 vesicle, *CTSA*, *CTSB*, *CTSD* coding for cathepsins A, B, and D; Supplementary Data 7) and genes involved in MHC class II antigen processing (*LGMN*, *IFI30*) suggesting that these genes are critical for the class II MHC restricted Env-specific response. Activation of the IRF7 innate antiviral program (*IFIH1*[9], *IRF7*; Supplementary Fig. 2) was also associated with vaccine-conferred protection of HIV-1 acquisition (Supplementary Fig. 2 and Supplementary Data 6). We also observed the enrichment of putative binding sites for IRF7 in the promoter regions of genes associated with a reduced risk of HIV-1 acquisition (Supplementary Data 6, 12/53 of IFNγ response genes identified in the transcriptomic analysis are also part of the geneset V$IRF7_01. This geneset includes genes that have a putative IRF7 binding site within ±2000 base pairs of their transcriptional starting sites), genes transcriptionally repressed in IRF7 siRNA transfection experiments[12] and genes induced by the overexpression of IRF7[13,14] (Supplementary Data 6). These results highlight the role of IRF7 as a

transcriptional regulator of the reduced risk of HIV-1 acquisition conferred by the RV144 vaccine.

While the IFNγ response pathway was associated with reduced risk of acquisition in vaccinees, the NF-κB and mTORC1 signaling pathways were associated with increased risk of HIV-1 acquisition in both vaccine and placebo recipients (Fig. 3). Genes implicated in NF-κB signaling that were associated with an increased risk of HIV-1 acquisition included genes of the NF-κB complex (*NFKB1*, *RELA*, *RELB*) and activators of the NF-κB complex (*BMP2*, *RIPK2*) (Supplementary Fig. 3 and Supplementary Data 7). Genes implicated in mTORC1 signaling that were associated with an increased risk of HIV-1 acquisition included regulators of the mTORC1 complex (*CXCR4*, *DDIT4*, *NAMPT*, *XBP1*) and its downstream targets (*CDKN1A*[15], *SLC2A3*[16], *TFRC*[17]). These results suggest that induction of IFNγ response genes is a vaccine-induced correlate of reduced risk of HIV-1 acquisition while induction of NF-κB related and mTORC1-related genes are vaccine-independent mechanisms associated with increased risk of HIV-1 acquisition.

**IFNγ pathway associated with HIV-specific antibodies and CD4s.** The six antibody and cellular assays which were previously assessed for their association with HIV-1 risk in RV144 vaccinees[2] as well as cell counts for 7 cell subsets measured by flow cytometry (FCM), 7 Luminex markers, 6 intracellular cytokine staining (ICS), and 31 MHC class II alleles measurements[4] were included in an integrative analysis (Supplementary Data 8). A projection-based approach that minimizes the technical effect specific to each dataset by generating a unique scale (i.e., projecting) for every dataset allowing to assess the correlation between different datasets was used for the integrative analysis[18]. This integrative analysis revealed that the IFNγ signaling pathway was significantly positively correlated to the frequency of Env-specific CD4$^+$ T cells (including IFNγ producing T cells) and to titters of IgG against V1/V2, a described correlate of low risk of HIV-1 acquisition in RV144 vaccinees (Supplementary Fig. 4). The IFNγ signaling pathway was also associated (through

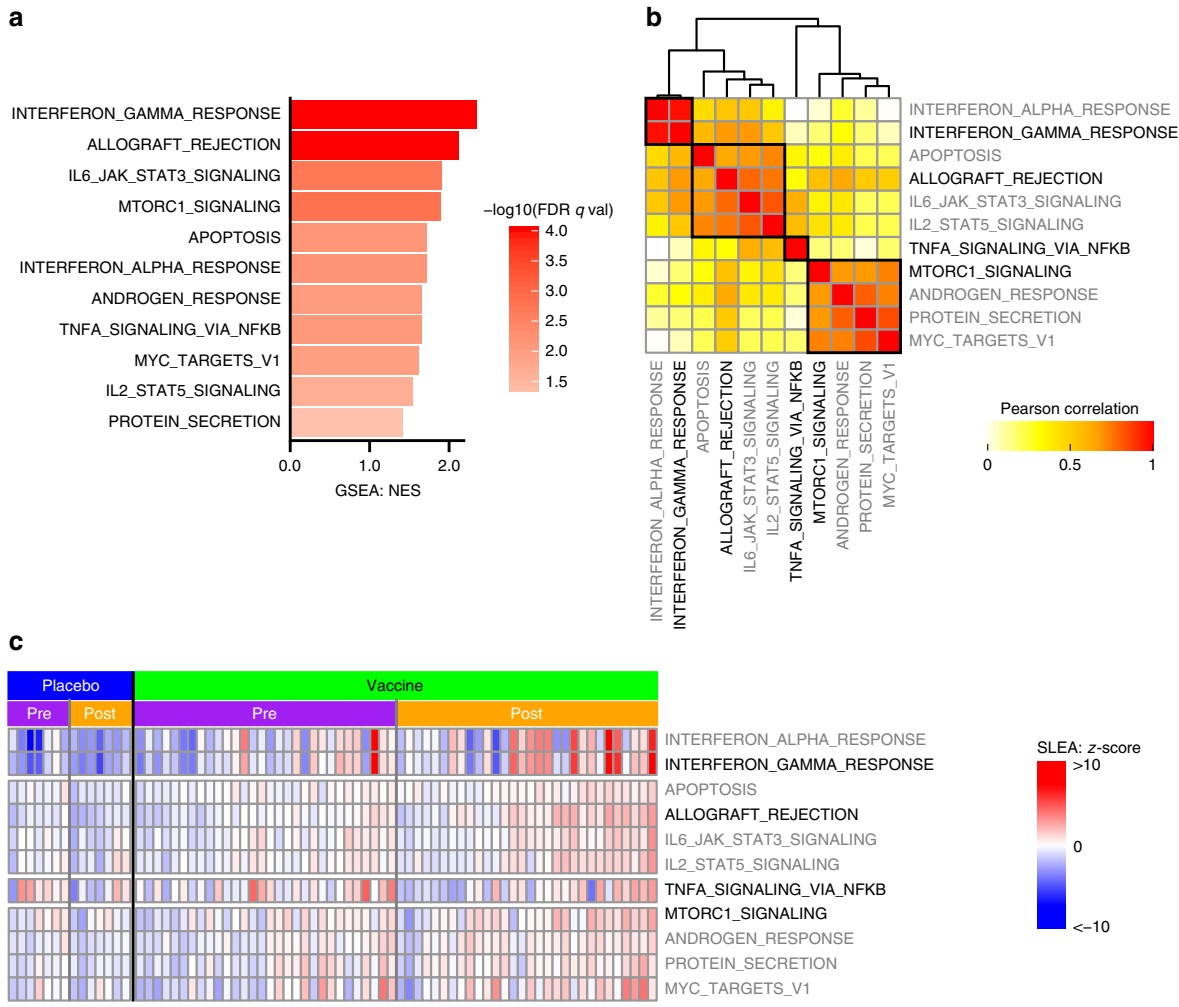

**Fig. 2** IFNγ response is strongly induced by the in RV144 vaccine. **a** Barplot presenting the pathways modulated by the RV144 vaccine two weeks after the last immunization compared to preimmunization. A normalized enrichment score (NES) greater than 0 corresponds to a pathway for which member genes are upregulated in vaccinees. Eleven pathways were significantly modulated after immunization in RV144 vaccinees but not in placebo recipients (DB: Hallmark, GSEA: FDR ≤ 5%). **b** Sample-enrichment analysis (SLEA) of those 11 pathways followed by clustering revealed that those pathways could be separated into four groups of highly correlated pathways (indicated by the black boxes). The representative pathway of each of the four groups (the most significantly enriched) is indicated in black while the remaining pathways are labeled in gray. **c** Heatmap presenting the SLEA z-score of each of the 11 pathways among the 40 vaccinees and 10 placebo recipients included in the transcriptomic pilot study at both timepoints investigated (pre: prevaccination, post: 2 weeks after the last immunization). An SLEA z-score greater than 0 corresponds to a pathway for which member genes are up-regulated while an SLEA z-score inferior to 0 corresponds to a pathway with genes downregulated in that sample

Env-specific CD4⁺ T cells) to the frequency of pDCs, the primary source of type I interferons (interferons α/β) in blood as well as to the PFS, another previously described correlate of low risk in RV144 vaccinees that integrates the cytokine response of CD4⁺ T cells in response to HIV-1 Env. The integrative analysis highlighted the relevance of the pathway identified above as they show their association with previously identified correlates of risk that underlie the major effector pathways of the immune response (IgG against V1/V2 and PFS).

This integrative analysis also revealed that the NF-κB pathway was significantly correlated with a higher frequency of monocytes and with heightened levels of the pro-inflammatory cytokines IL2 and IL3 (Supplementary Fig. 4). To further investigate the association between the NF-κB pathway and specific cell subsets including monocytes, we performed a deconvolution[19] (i.e., separation) of the PBMC gene expression profile into six major immune subsets (B cells, T cells, NK, monocytes, mDC, and pDC) using the frequencies of those subsets measured by FCM (see Methods and Supplementary Fig. 5a). Analysis of the

deconvoluted gene-expression profiles revealed that genes of the NF-kB pathway were expressed at significantly higher levels in monocytes compared to the other five subsets (Supplementary Fig. 5b, Wilcoxon rank-sum test: $p = 1.64e−06$), thereby supporting the results of the integrative analysis (Supplementary Fig. 4).

**The mTORC1 pathway is a marker of HIV-1 acquisition.** To test if the IFNγ signaling pathway could be a novel marker of low infection risk in vaccinees, we built a logistic regression model combining gender, behavior risk and previously described correlates of risk (IgA against Env, IgG against V1/V2, DQB1*06 allele, DPB1*13 allele, and PFS) to predict acquisition of HIV-1 among RV144 vaccinees. This model was then compared to one that included results from gene-expression profiling. The balanced accuracy of both models was assessed by tenfold cross-validation. The best model built without gene expression showed a balanced accuracy of 62.3%, while the best model that included results from gene expression had a balanced accuracy of

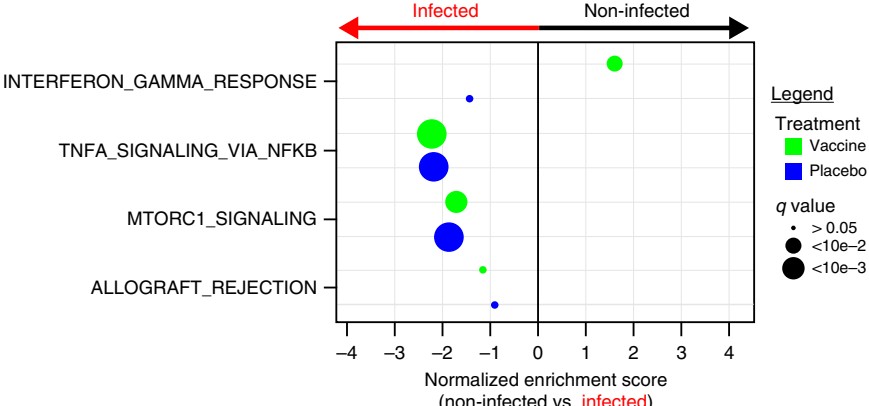

**Fig. 3** IFNγ response associated with the reduction of the risk of HIV-1 infection in vaccinees. Dotplot presenting the association between the pathways induced by the RV144 vaccine and HIV-1 infection status, separately for vaccinees and placebo recipients. Gene-expression of 183 vaccine recipients, 30 cases and 153 controls, and 30 placebo recipients, of which 17 were infected, were used for this analysis. GSEA was performed and identified one pathway associated with the reduction of the risk of HIV-1 acquisition in vaccinees and the two pathways associated with a higher risk of HIV-1 acquisition both in vaccinees and placebo recipients. The normalized enrichment scores (NES) of those pathways are presented on the plot. An NES greater than 0 suggests that participants with higher expression of the genes in that pathway are less likely to be infected by HIV-1 while an NES below 0 corresponds to participants with higher expression of the genes in that pathway and more likely to acquire HIV-1. The size of the dots is proportional to the false-discovery rate (q value) of the enrichment

67.9% (Fig. 4a). Receiver operating characteristic analysis was performed to compare these two models; no statistically significant gain in accuracy resulted from the addition of gene-expression results to previously described correlates (Fig. 4b). A multivariate logistic regression was built using all the candidate markers of protection to assess their relative contribution to the prediction of HIV-1 acquisition among RV144 vaccinees (Table 1). Only the PFS, the interaction term IgA:DQB1*06, the interaction term IgG:DPB1*13 (i.e., association between IgG level with HIV-1 acquisition separately for DPB1*13− and DPB1*13+ vaccinees) and mTORC1 signaling remained statistically significantly associated to HIV-1 acquisition in a multivariate model. The IFNγ pathway identified by the transcriptional profiling (univariate analysis: Odd ratio = 0.883 $p$ = 0.00837; Supplementary Table 6) did not bring an independent contribution to the prediction of the risk of HIV-1 acquisition among RV144 vaccinees (multivariate analysis: Odd ratio = 0.974 $p$ = 0.677; Table 1). This analysis suggests that the IFNγ pathway identified by gene expression was likely confounded (i.e., bring similar predictive information) with the cellular (PFS) and serological/genetic (IgG:DPB1*13) correlates of the RV144 vaccine-conferred low risk of HIV-1 acquisition (Supplementary Table 6).

We then assessed whether the IFNγ pathway was confounded with IgG:DPB1*13. Thus, we stratified the RV144 vaccinees by the DPB1*13 allele and evaluated the association between the IFNγ pathway and IgG antibodies binding to V1/V2. Stratifying RV144 vaccinees by the DPB1*13 allele revealed that the IFNγ signaling pathway and IgG antibodies binding to V1/V2 were positively correlated to each other only in DPB1*13+ vaccinees (Fig. 5a, b and Supplementary Data 9–12); moreover, the association of IFNγ signaling pathway with low risk was more pronounced in DPB1*13+ vaccinees (Fig. 5c). Conversely, the IFNγ signaling pathway was not correlated to IgG antibodies binding to V1/V2 nor HIV-1 acquisition in DPB1*13− vaccinees. A significant overlap of 38 genes was observed between IFNγ response genes correlated with IgG antibodies binding to V1/V2 in DPB1*13+ vaccinees (103 IFNγ response genes) and IFNγ response genes negatively associated with HIV-1 acquisition (53 IFNγ genes; Fisher's exact test: $p$ = 0.00367). Those 38 genes included the transcription factor IRF7 and its target genes known to block viral entry (XCL1) or prevent HIV-1 virion assembly in

infected cells (IFITM3, ISG15, MX2, TRIM26). Several genes encoding components of the killing machinery required for ADCC function, i.e., CASP3[20], FAS[21], TNFSF10[21] (Supplementary Data 12) were included among the genes that correlated with titers of IgG antibodies binding to V1/V2 suggesting that ADCC responses could correlate with the decreased rate of virus acquisition after vaccination. Concomitantly, we observed within the IRF7 signature the expression of genes that could be involved in dampening Th1 cell development (IL18BP) or in driving the response toward Th2 cells (PARP14). Genes with a potent anti-inflammatory activity that can suppress global immune activation (SERPING1, LY6E) were also expressed in the protective transcriptomic signature. ITGB7, an integrin that is expressed by T cells and NK cells known for homing the gut was included in the IRF7 target genes that were associated to protection from simian immunodeficiency virus (HIV-like virus) acquisition[22]. While several of the 38 genes included in the IRF7 signature were known to be regulated by IFNγ, 24 out of 38 were also IFNα stimulated genes (Supplementary Data 11). In vitro experiments performed on healthy blood samples confirmed that IRF7 phosphorylation (measured by FCM) was induced by more than 1.5-fold upon treatment with interferons (IFNα, IFNβ, or IFNγ). Interferon treatment rendered host cells on average eight times more resistant to in vitro HIV-1 infection (Fig. 5d, e and Supplementary Fig. 6). These results show for the first time the possible contribution of innate antiviral immune responses to lower the risk of HIV-1 acquisition among RV144 vaccinees and similar to what was previously reported by Haynes et al.[2], they highlight the contribution of innate cellular functions (NK/ADCC) as correlates of risk in RV144, suggesting that these innate immune functions may play an essential role as correlates of RV144 vaccine-protection.

The IFNγ signaling pathway that correlated with PFS, a correlate of reduced risk of HIV-1 acquisition distinct from IgG antibodies against V1/V2, included a different set of genes with the transcription factor STAT1 as their key regulator (Supplementary Fig. 7 and Supplementary Data 13–15). Several genes involved in the upregulation of MHC class I (TAP1, TAPBP) and class II antigen presentation (HLA-DRB1, HLA-DMA), the initial step in the priming of T cell responses, as well as genes involved in the development of helper T cell functions (IL15, IL15RA,

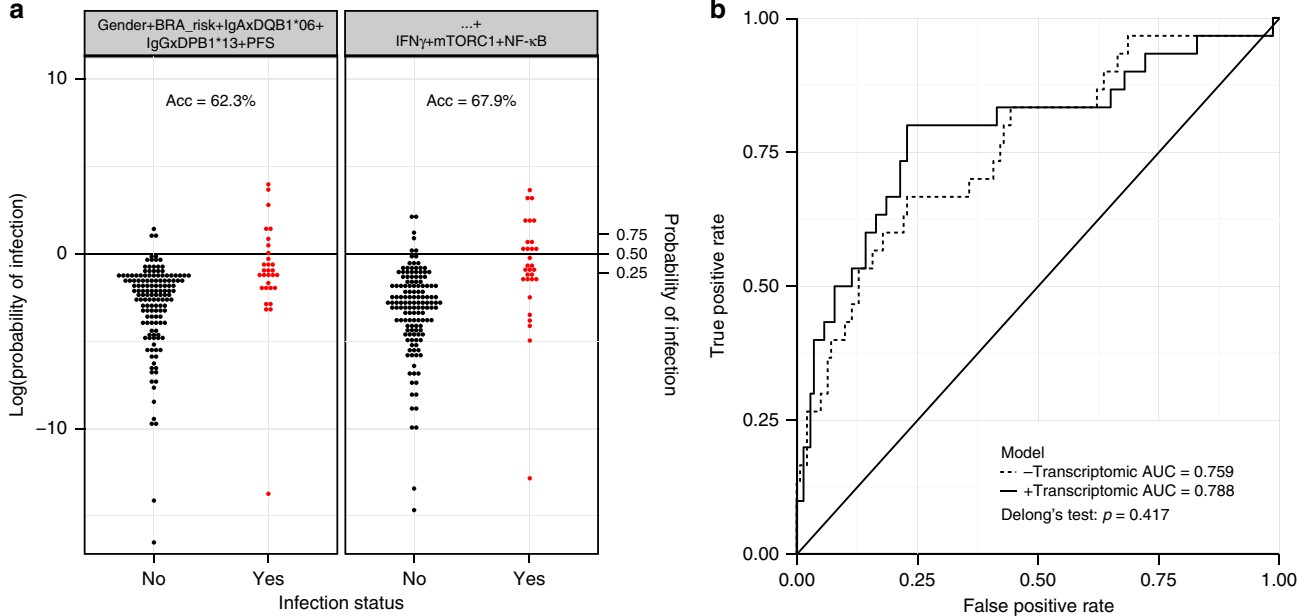

**Fig. 4** Prediction of the response does not improve by adding transcriptomic data. **a** Logistic regression models were built to predict HIV infection status of RV144 vaccinees (142 vaccinees that were HIV negative at last follow-up and 30 vaccinees that acquired HIV). The accuracy of each model was assessed by tenfold cross-validation. The first model (left panel) included IgA against V2, IgG against V1/V2, and the polyfunctionality score (PFS) previously identified as markers of response to RV144 vaccine. The second model (right panel) included the same markers as the first model but with the addition of the three pathways associated with HIV status in the transcriptomic analysis (IFNγ response, MTORC1 signaling, and TNFα signaling via NF-κB). The balanced accuracy (Acc) of each model is given on the plot. **b** Corresponding ROC curves based 10-fold cross-validation for the model without the three genesets and the model with the three genesets identified in the transcriptomic analysis

**Table 1 Univariate and multivariate analysis of markers of HIV-1 acquisition among vaccinees**

|  | Univ. OR (95% CI) | Univ. p | Multiv. OR (95% CI) | Multiv. p |
|---|---|---|---|---|
| IgA antibodies binding to Env | 1.54 [1.06, 2.26] | **0.0237** | 1.22 [0.733, 2.02] | 0.432 |
| IgG antibodies binding to V1/V2 | 0.703 [0.446, 1.07] | 0.112 | 1.24 [0.670, 2.29] | 0.485 |
| DQB1*06 | 1.12 [0.408, 2.78] | 0.819 | 0.505 [0.0297, 2.75] | 0.517 |
| Interaction IgA:DQB1*06 | – | – | 9.77 [1.95, 112] | **0.0216** |
| DPB1*13 | 0.696 [0.270, 1.64] | 0.426 | 0.453 [0.0975, 1.53] | 0.243 |
| Interaction IgG:DPB1*13 | – | – | 0.132 [0.0216, 0.537] | **0.0115** |
| PFS | 0.620 [0.390, 0.944] | **0.0322** | 0.477 [0.252, 0.844] | **0.0153** |
| INTERFERON_GAMMA_RESPONSE | 0.883 [0.803, 0.967] | **0.00837** | 0.974 [0.857, 1.10] | 0.677 |
| MTORC1_SIGNALING | 1.33 [1.15, 1.57] | **0.000336** | 1.25 [1.01, 1.56] | **0.0496** |
| TNFA_SIGNALING_VIA_NFKB | 1.24 [1.10, 1.42] | **0.000871** | 1.08 [0.919, 1.28] | 0.368 |

For each variable, the odds ratio (OR) and its 95% confidence interval (CI) is reported per one standard deviation increase. The p value of a z-test testing that the OR is different from 1 is reported in the table. p values inferior or equal 0.05 are indicated in bold. All univariate (univ.) and multivariate (multiv.) logistic regression models were adjusted for gender and behavior risk of the participants

*IL4R*) were involved in the positive correlation observed between the STAT1 target genes part of the IFNγ pathway and the PFS score (Supplementary Data 15).

TNFα signaling via NF-κB (41 genes) and mTORC1 (39 genes) signaling pathways were not associated with previously identified correlates of the RV144 vaccine response. Herein, we provide evidence that these two pathways were associated with an increased risk of HIV-1 acquisition both in placebo- and vaccine-recipients (Fig. 6 and Supplementary Data 5). The NF-κB signaling pathway included markers of activated T cells and their survival (*BCL2A1*, *IL12B*, *TNFSF9*), cell migration (*EFNA1*, *CXCL2*, *CCL20*) and induction of proinflammatory prostaglandins (*PTGS2*). The mTORC1 pathway, with BHLHE40 as the upstream transcription factor, included genes known to be important for HIV-1 entry into target cells (co-receptor for HIV-1 on CD4+ T cells *CXCR4*, *SLC2A1*, *UNG*) as well as genes involved in HIV replication (*ETF1*, *HMGCS1*, *PGM1*). Both mTORC1 and NF-κB

pathways included genes that were positive regulators of cell cycle (*CCDN1*, *CCNG1*), genes that can inhibit cell cycle progression and downstream of the immune suppressive TGF-beta signaling pathway (*TGIF1*, *PPP1R15A*). These results suggest that controlling the balance between pro- and anti-inflammatory pathways trigger the development of protective vaccine responses.

## Discussion

Transcriptional profiling of PBMCs from RV144 vaccinees stimulated with Env peptides was characterized by the upregulated expression of genes associated with antigen presentation, maturation of MHC class II complex, and genes endowed with antiviral functions; these genes and pathways were induced only in vaccinees that remained HIV-1 negative at their last follow up (control) compared to vaccinees that acquired HIV-1 (cases). Induction of these pathways was not observed in placebo

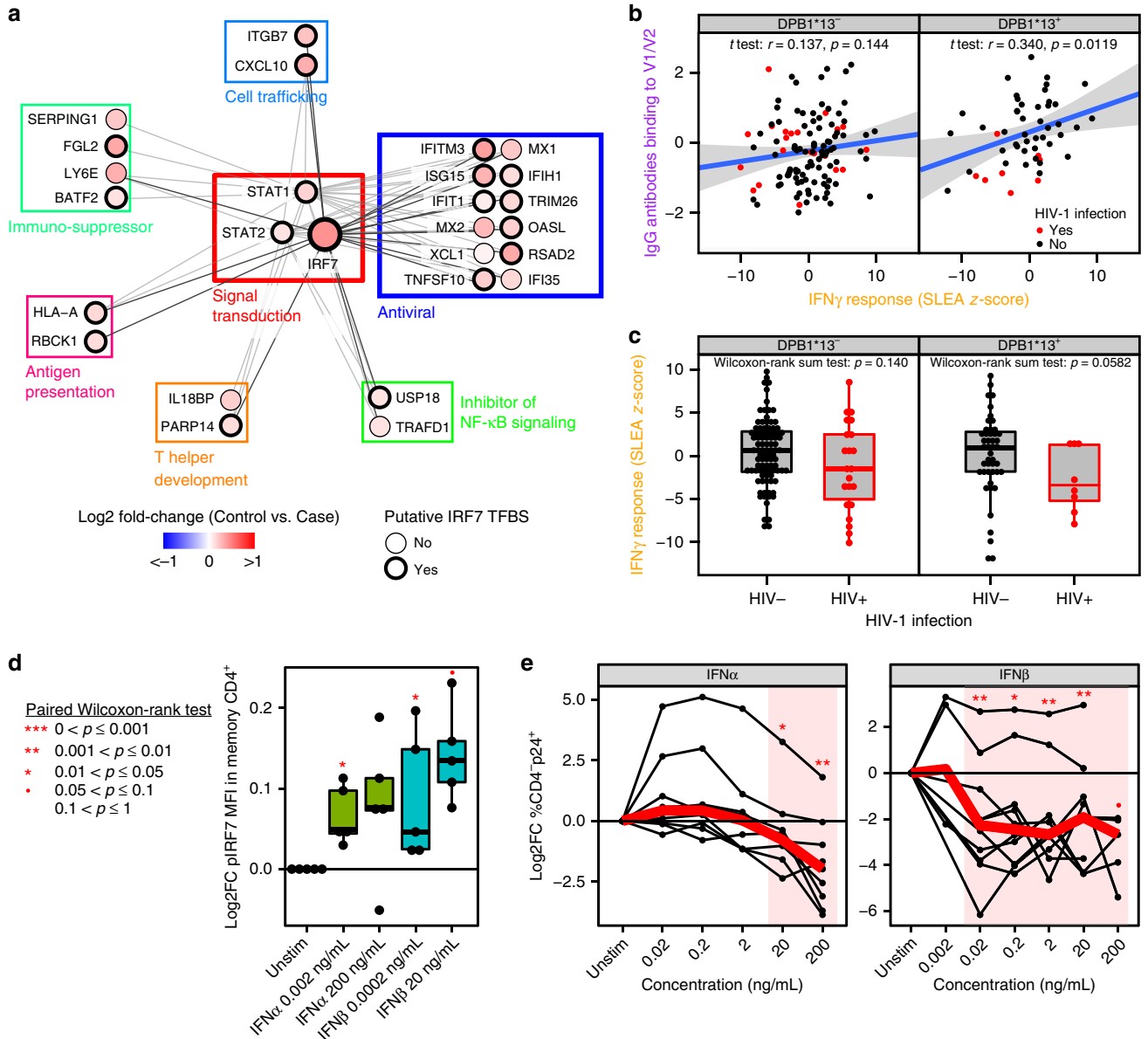

**Fig. 5** Mechanisms associated with a reduced risk of HIV-1 acquisition among RV144 vaccinees. **a** Network showing the genes implicated in IFNγ signaling with annotated functions. Nodes correspond to genes; the color of a node is proportional to the log2 fold-change between controls and HIV-1 cases. Edges are inferred by GeneMANIA and correspond to physical interactions, colocalization, or co-expression. The remaining genes part of this signature but with unknown/unrelated functions can be found in Supplementary Data 5. **b** Scatter plot presenting the expression of IFNγ responsive genes as a function of the levels of IgG antibodies binding to V1/V2 and DPB1*13 alleles. The average expression of the IFNγ genes was calculated using the SLEA z-score method. A linear regression model (blue line), and its 95% confidence interval (gray zone), was fit between SLEA z-score and IgG antibodies against V1/V2, and this separately for each DPB1*13 allele. A Pearson correlation and a *t* test were performed to assess the significance of the correlation between the transcriptomic data and antibody response. **c** Scatter plot presenting the association of IFNγ target genes and HIV-1 acquisition, separately for patients DPB1*13− and DPB1*13+. Wilcoxon-rank sum test was performed to assess the significance of the association between the transcriptomic data and HIV-1 acquisition. **d** Boxplot of the ratio of phosphorylated IRF7 in memory CD4+ cells stimulated with interferon compared to unstimulated memory CD4+ cells. The ex vivo experiments were performed on cells from five healthy donors. The fold-change in the median fluorescence intensity (MFI) between interferon stimulated samples and the unstimulated condition is presented as a function of the concentration of interferon α and β used. **e** Lines plot showing the ratio of the frequency of CD4−p24+ after interferon stimulation over the unstimulated levels as a function of interferon concentration. The red lines indicate the median frequencies of CD4−p24+ across ten healthy donors. **d, e** A paired Wilcoxon rank-sum test was used to assess the statistical significance of the fold-change (***$p \leq 0.001$, **$0.001 < p \leq 0.01$, *$0.01 < p \leq 0.05$, •$0.05 < p \leq 0.1$)

recipients or in the absence of stimulation with Env peptides indicating that the vaccine triggered those pathways. Ex vivo experiments showed that IRF7 (a key regulator of an antiviral innate immune response) associated with low risk in vaccinees, was expressed by T cells. Expression of IRF7, and more

importantly genes with an antiviral activity that are regulated by IRF7 (*XCL1*, *IFITM3*, *ISG15*, *MX2*, *TRIM26*), in T cells can render these cells less susceptible to HIV-1 infection[23–28]. These results provide a mechanism whereby CD4+ T cells from subjects immunized with the RV144 vaccine will mount an Env specific

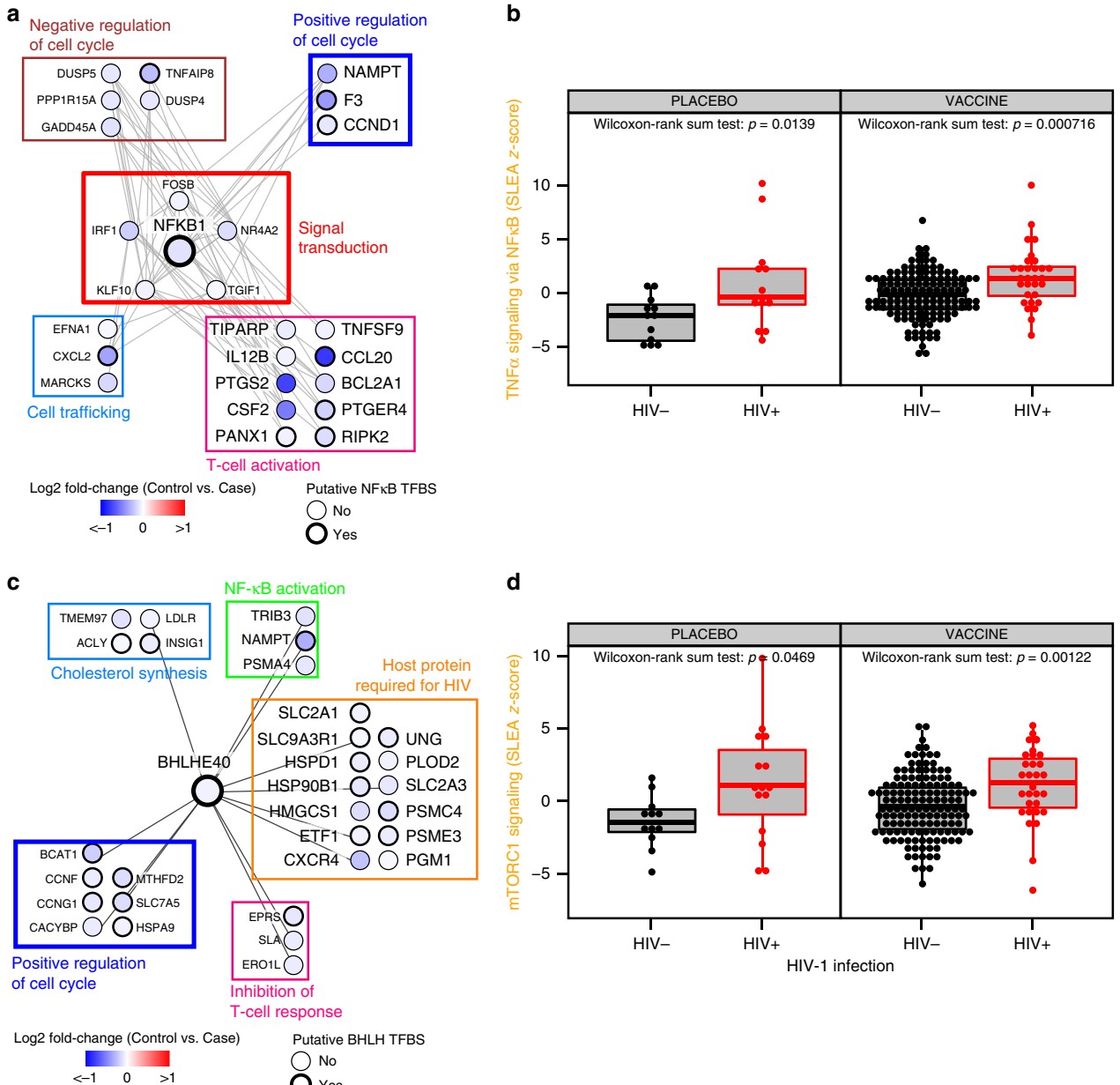

**Fig. 6** Mechanisms associated with increased risk of HIV-1 acquisition. **a** Network showing the genes implicated in NF-κB signaling. Nodes correspond to genes; the color of a node is proportional to the log2 fold-change between controls and HIV-1 cases. Edges are inferred by GeneMANIA and correspond to physical interactions, colocalization. or co-expression. The remaining genes part of this signature but with unknown/unrelated functions can be found in Supplementary Data 5. **b** Boxplot presenting the association of genes implicated in NF-κB signaling and HIV-1 acquisition, separately for placebo recipients and vaccinees. Wilcoxon-rank sum test was performed to assess the significance of the association between the transcriptomic data and HIV-1 acquisition. On the boxplot, the lower whisker, the lower hinge, the midhinge, the upper hinge and the upper whisker correspond to the interquartile (IQR) from the first quartile, the first quartile, the median, the third quartile and the IQR from the third quartile, respectively. **c** Network showing the genes implicated in mTORC1 signaling. Nodes correspond to genes; the color of a node is proportional to the log2 fold-change between controls and HIV-1 cases. Edges are inferred by GeneMANIA and correspond to physical interactions, colocalization, or co-expression. The remaining genes part of this signature but with unknown/unrelated functions can be found in Supplementary Data 5. **d** Boxplot presenting the association of genes implicated in mTORC1 signaling and HIV-1 acquisition, separately for placebo recipients and vaccinees. Wilcoxon-rank sum test was performed to assess the significance of the association between the transcriptomic data and HIV-1 acquisition

type II interferon (interferon γ) response that could, in turn, trigger the expression of these antiviral genes (e.g., *XCL1*, *IFITM3*, *ISG15*, *MX2*, *TRIM26*), thereby rendering these cells and bystander cells resistant to HIV-1 infection. Attenuated viruses which are all known to be very efficacious vaccines (YF17D, measles, smallpox) are known to trigger these innate antiviral immune response pathways[29,30].

Integration of gene expression, antibody responses, and ICS datasets revealed that the IFNγ pathway correlated with IgG antibodies binding to V1/V2 (explaining 5% of the variance of the IFNγ pathway) and with cytokine production by Env-specific CD4+ T cells (explaining 8% of the variance of the IFNγ pathway). Herein, we show that different signatures can independently predict immunogenicity of the vaccine (STAT1 regulated

genes) and low risk of acquisition (IRF7 target genes) confirming the nonredundant roles of IgG antibodies binding to V1/V2 and T cells polyfunctionality. Indeed, antiviral genes regulated by IRF7 were significantly associated with IgG response while STAT1-target genes implicated in antigen presentation via MHC class I and class II were specifically correlated with the PFS. The IRF7 signature was enriched in IFNα-stimulated genes (24/38 genes) compared to the STAT1 signature (34/97 genes, $X^2$ test: $p = 0.00555$) suggesting that different stimuli triggered the IRF7 and the STAT1 gene signatures; type I interferons stimulated the IRF7 gene signature, while type II interferons could specifically induce the STAT1 gene signature. In support of our observations, knock out of IRF7 in murine models did not abrogate CD4+ T cell responses[31] while it led to enhanced viral replication. In contrast, knock out of the STAT1 gene in mice did abrogate antigen-specific CD4+ T cell responses. In line with these reports, STAT1 regulated genes were specifically correlated to the heightened polyfunctional CD4+ T cell response linked to low risk in RV144 vaccinees and not to a general "non-protective" T-cell response (e.g., monofunctional CD4+ T cells that can secrete only one cytokine). The IRF7 signature was specifically correlated to an IgG response linked to low risk in RV144 vaccinees and was not correlated to other "non-protective" antibody markers such as IgG response to non-V1/V2 epitopes. Further evidence suggests that both IRF7 and STAT1 activation are required for the development of the protective immune response also supported by the fact that knock-out of STAT1 and induction of IRF7 is lethal in virally infected animal models as it leads to an uncontrolled cytokine storm[32]. Our data shows that the antiviral innate immune response and the HIV specific CD4+ T-cell response are two independent correlates of low risk of HIV-1 acquisition.

The IRF7 antiviral transcriptomic program and the IgG antibodies binding to V1/V2 were heightened in vaccinees that express DPB1*13 allele. This association may result from a T-cell response to specific T-cell epitopes present in Env antigen and restricted by MHC class II of DPB1*13. The protective role of DPB1*13 could also be attributed to the poor IgA responses previously shown to occur in DPB1*13+ subjects[33]. IgA antibodies compete with IgG antibodies for binding to HIV-1 Env and thus abrogate ADCC in vaccinees[6]. IgG responses have been shown to mediate antibody-dependent cellular cytotoxicity ADCC[6]. Type I interferon (including IRF7) induces the expression of FCGR that triggers ADCC; the latter has been suggested to be a mechanism of RV144 vaccine-mediated protection[2]. Binding of IgG antibodies (specific for V1/V2) to FCGR will trigger ADCC and will induce IRF7 and type I interferons[34]. Gene-expression analysis confirmed the potential role of ADCC in response to the RV144 vaccine since several genes implicated in ADCC are induced by the vaccine; they include markers of NK cells (CD48, KIR2DL1, KIR3DL1, KIR3DL2, NKG2C, FCGR3A) as well as effector molecules of ADCC (NCR1, NCR2, FAS, GZMB, PRF1, TNFSF10); of note only FCGR3A, TNFSF10 were associated with a reduced risk of HIV-1 acquisition. In addition, IRF7-induced genes important for cell trafficking (e.g., CXCL10 and ITGB7) may promote migration of effector cells to the mucosal sites where HIV-1 infection will occur. Of note, ITGB7 was confirmed in an animal model as a correlate of protection of an RV144-like vaccine[22].

Integration of all these datasets suggests a model whereby ALVAC vector, used to prime RV144 vaccinees and know to be able to infect dendritic cells, trigger pDCs to produce type I interferons as ALVAC interact the innate sensor STING[35]. Type I interferon will upregulate antiviral genes in bystanders cells including Env-specific activated CD4 T cells. These cells now can provide help to B cells to produce IgG and to other CD4 T cells.

This model can be deduced only by integrating multiple OMICs and will be validated in subsequent clinical trials.

Our analysis indicated that genes downstream of the proinflammatory transcription factor NF-κB as well as genes downstream of mTORC1 that are required for HIV-1 life-cycle (CXCR4, ETF1[36]) were associated with the risk of HIV-1 acquisition; those associations were observed in the placebo and vaccine arms of the study and thus were independent of vaccination. Hence, participants expressing the mTORC1 and NF-κB signatures did not benefit from the RV144 vaccine. Moreover, those results are not supportive of vaccine-related enhancement of HIV-1 acquisition in RV144 vaccinees.

Both the integrative analysis between gene expression and frequency of cells measured by FCM as well as in the deconvolution of the gene expression confirmed monocytes as the cellular subset that expressed high levels of NF-κB and its downstream targets. Activated monocytes that express high levels of NF-κB genes would trigger the production of proinflammatory cytokines/chemokines such as IL2 and IL3 that can enhance the survival of activated T cells thereby providing HIV-1 with potential target cells to infect and lead to HIV-1 acquisition susceptibility. Moreover, the association with increased acquisition of genes that regulate TGFβ signaling or genes that are downstream of the antiproliferative cytokine TGFβ could result from the immunosuppressive activity of TGFβ on the development of protective HIV-1 specific CD4+ or CD8+T cell responses[36]. TGFβ is also known to regulate IgA class switch; IgA against Env was shown to be associated with a higher risk of HIV-1 acquisition2.

In addition, host proteins required for HIV-1 life cycle downstream of mTORC1 signaling pathways (CXCR4, ETF1[37]) were elevated in both placebo recipients and vaccinees. Rapamycin, that downregulates mTORC1 pathway, could improve the effectiveness of the RV144 vaccine as it did enhance the response to flu vaccination in a cohort of elderly subjects[38].

Our study highlights the important contributions of unbiased system biology approaches in defining mechanisms underlying vaccine-mediated protection. Similar approaches could lead to the identification of host-related markers associated with vaccine-conferred protection by investigating prevaccination gene-expression profiling of participants receiving the RV144 vaccine. Moreover, the contribution of the mucosal immune response needs to be assessed since we have previously shown using similar unbiased approaches that integrate mucosal and systemic immune responses can inform us on mechanisms leading to vaccine-conferred protection[22]. Differential gene expression could also result from polymorphisms in the coding and regulatory regions of those genes, alternative splicing, chromatin accessibility or noncoding RNA expression[39]. The platform used in this study, microarrays, does not provide us with the ability to investigate all those regulatory elements. Further studies using such high-dimensional data types would further complement the mechanistic insights identified in this study that lead to the vaccine-conferred protection by the RV144 vaccine. Finally, follow-up studies using in vitro experiments and animal models are required to confirm the functional/mechanistic contribution of the aforementioned pathways.

In conclusion, we have shown that the establishment of a productive HIV-1 infection in participants depends on the balance between innate antiviral and proinflammatory responses. The proinflammatory responses mediated by mTORC1 and NF-κB signaling can lead to the activation and proliferation of HIV-1 target cells. Immune modulators that boost innate antiviral responses and suppress proinflammatory detrimental immune responses may decrease the risk of HIV-1 infection or replication. The HVTN 702 trial, the follow-up efficacy trial of a pox-protein

vaccine regimen initiated in South Africa in Q4 2016, will allow us to evaluate and confirm the mechanisms identified in this study as being associated with the RV144 correlates.

## Methods

**Study design**. Fifty participants of the RV144 clinical trial were part of the transcriptomic pilot study, randomly sampled within each (gender × treatment arm) strata (50% for each gender, 80% vaccine recipients) among subjects completing follow-up HIV negative. This ensured that baseline characteristics of subjects enrolled in the transcriptomic pilot study were similar to the original RV144 cohort (16,402 participants) except for greater proportions of vaccinees and participants that completed the trial protocol in the transcriptomic pilot study cohort (Supplementary Table 1). Separately, 183 participants of the RV144 clinical trial were selected for the transcriptomic case/control study. Baseline characteristics of subjects enrolled in the transcriptomic case/control study were similar to the original RV144 cohort (16,402 participants) except for greater proportions of vaccinees, participants that completed the trial protocol and participants that acquired HIV-1 in the transcriptomic pilot study cohort (Supplementary Table 3). No imbalance was observed in term of clinicopathological characteristics between the vaccinees and placebo recipients included in the transcriptomic pilot study cohort (Supplementary Table 2) or the transcriptomic case/control study cohort (Supplementary Table 4). The RV144 trial protocol was reviewed by the ethics committees of the Ministry of Public Health, the Royal Thai Army, Mahidol University, and the Human Subjects Research Review Board of the U.S. Army Medical Research and Materiel Command. All participants gave their informed consent. Written informed consent was obtained from all volunteers.

**Vaccine**. ALVAC-HIV (vCP1521) is a recombinant canarypox genetically engineered to express HIV-1 gag and pro (subtype B, LAI strain) and CRF01_AE (subtype E) HIV-1 gp120 (92TH023) linked to the transmembrane 3 anchoring portion of gp41 (LAI). AIDSVAX B/E is an HIV gp120 envelope glycoprotein vaccine containing a subtype E envelope from the HIV-1 strain A244 (CM244) and a subtype B envelope from the HIV-1 MN. The envelope glycoproteins, 300 μg of each, are co-formulated with 600 μg of alum adjuvant. ALVAC-HIV placebo consisted of virus stabilizer and freeze-drying medium in 1 ml sodium chloride. AIDSVAX placebo was 600 μg alum adjuvant.

**Primary endpoint**. HIV infection was monitored every 6 months, from month 6 to month 36 after the initial immunization. HIV infection established from repeated positive results on enzyme immunoassay and Western blots, with two confirmatory HIV nucleic acid tests: the Amplicor HIV Monitor (version 1.5) assay (Roche) in Thailand and the Procleix HIV discriminatory assay (Novartis) in the United States. Correlates analyses defined the primary endpoint as the diagnosis of HIV-1 infection any time after the month six visit postinitial immunization.

**Transcriptomic analysis**. PBMC samples taken preimmunization and 2 weeks (window: −2 to +14 days) after the last immunization, were either stimulated in vitro for 15 h with Env peptides or with the vehicle (dimethyl sulfoxide). The Env peptides consisted of 15 amino acids spanning the Env 92TH023 sequence expressed in vCP1521 and overlapping by 11 amino acids (Biosynthesis, Lewisville, TX) were combined into one pool at a final concentration of 1 μg/ml per peptide, and used to stimulate $10^6$ PBMC ex vivo; as further detailed in Haynes et al.[2]. The transcriptomic profile of the stimulated PBMC was assessed using Illumina Human HT-12 beadchips. RNA was isolated using the Rneasy micro kit (Qiagen) and the quantity and quality of the RNA were confirmed using a NanoDrop 2000c (Thermo Fisher Scientific) and an Experion Electrophoresis System. Samples (50 ng) were amplified using Illumina TotalPrep RNA amplification kits (Ambion). The microarray analysis was conducted using 750 ng of biotinylated complementary RNA hybridized to Human HT-12 version 4 beadchips (Illumina) at 58 °C for 20 h. The chips were scanned using Illumina's iSCAN and quantified using Genome Studio (Illumina).

Raw beadchips intensities were quantile-normalized and log2-transformed. The LIMMA framework was used to fit linear regression model with the log2 gene expression as dependent variable and the groups of interest (vaccination group, HIV-1 infection status, or antibody response) as independent variables in order to identify genes differentially expressed between vaccination group (vaccine or placebo), HIV-1 infection status (control versus case) or genes correlated to antibody response (IgG antibodies binding to V1/V2). A moderated t test was used to assess the statistical significance of the association between gene expression and the groups of interest. Benjamini and Hochberg correction was applied to adjust for multiple testing.

Genecards[40], Reactome[41], GeneRIF[42], and Literome[43] were used to annotate the function of genes. The HIV-1 host factors were obtained from the NCBI HIV-1 interaction database[44].

GSEA was used to identify pathways modulated after Env stimulation and/or associated with HIV-1 acquisition[45]. In GSEA, the most varying probe across samples was used as representative of redundant probes annotated to the same gene. The gene list ranked by LIMMA moderated t-statistic were used as input for

the GSEA analysis. The pathways (i.e., genesets) database used for all GSEA analysis were the Molecular Signatures Database (version 5.1) hallmark genesets[46], canonical pathways (module C2.CP), transcription factor targets (module C3.TFT), and blood cells markers[47]. The GSEA Java desktop program was downloaded from "http://www.broadinstitute.org/gsea/index.jsp [http://www.broadinstitute.org/gsea/index.jsp]" and the default parameters of GSEA preranked module (number of permutations: 1000; enrichment statistic: weighted; seed for permutation: 101, 15 ≤ gene set size ≤ 500) were applied for analyses. Putative transcription factor binding sites were identified in the regulatory region of genes associated with HIV-1 acquisition using HOMER version 4.9 using default parameters[48].

SLEA was used following GSEA analysis to investigate the enrichment of pathways in the different samples[49]. Briefly, the expression of all the genes in a specific pathway was averaged across samples and compared to the average expression of 1000 randomly generated genesets of the same size. The resulting z-score is then used to reflect the overall perturbation of a pathway in a sample.

**Intracellular cytokine staining**. PBMC were plated in a 96-well plate ($10^6$ cells per well). PBMC stimulation was performed in 10% FBS/RPMI media in the presence of 1 μg/ml anti-CD28 and anti-CD49d and Brefeldin A (BD Biosciences, San Diego, CA) and stimulated with HIV peptides (New England Peptide, Gardner, MA) of 15-mer overlapping by 11 amino acids representing HIV subtype E-Env (TH023; 162 peptides) and HIV subtype B-Gag (LAI; 120 peptides). PBMC supplemented with DMSO was used as a negative control. After 6 h of stimulation at 37 °C, 5% $CO_2$ EDTA (20 mM, Sigma) was added and incubated for 15 min. Subsequently, PBMC were fixed and permeabilized using FACS lysing solution and FACS permeabilizing solution 2 (BD) according to the manufacturer's instructions. The following antibodies were added for 60 min at room temperature in the dark: CD4-fluorescein isothiocyanate (FITC), CD3-allophycocyanin (APC), IFNγ-phycoerythrin (PE), Il-2-phycoerythrin (PE), and CD8-PerCP-Cy5.5 (all BD Biosciences). PBMC were washed and fixed with 1% paraformaldehyde. The analysis was performed using a FACSCalibur flow cytometer (BD Immunocytometry Systems). ICS data were provided to us by the trial investigators in either an ICS-positivity score (call) format and aggregate value format[2]. ICS analytes included CD154, IFNγ, IL-4, IL-2, IL-17α, and TNFα.

**Multiplex cytokine bead array**. Cryopreserved PBMC were thawed and rested overnight. Totally, $5 \times 10^5$ PBMC each were stimulated with Env 92TH023 peptides at 37 °C and 0.5% DMSO served as negative control. After 48 h, supernatants were harvested and frozen at −80 °C until analysis. Analyte concentrations were measured using a MILLIPLEX MAP Human Cytokine/Chemokine—Custom-12-Plex kit (Millipore, Billerica, MA) following instructions provided by the manufacturer. All samples were acquired on a Luminex 200 instrument (Millipore) and data analyses were performed using MasterPlex software. Multiplex cytokine (Luminex) data were provided to us by the trial investigators as normalized values (mean of 0, a standard deviation of 1)[2]. Luminex analytes included GM-CSF, IFNγ, IL-2, IL-3, IL-4, IL-5, IL-9, IL-10, IL-13, MIP1β, TNFα, and TNFβ.

**FCM phenotyping panel**. Samples were stained for extracellular markers for FACS analysis following the manufacturer's recommendations (Becton Dickinson Cytofix/Cytoperm Kit). Samples were surface stained with the following antibodies to distinguish cell subsets: CD11c (Pe-Cy5), CD14 (FITC), HLA-DR (Allophyco-cyanin-Cy7), CD3 (Qdot 800), CD19 (Qdot 605), and CD123 (Brilliant Violet). They were acquired using a Becton Dickinson LSR II flow cytometer and analyzed using FlowJo (TreeStar). Antibodies were from Becton Dickinson unless otherwise stated. The FCM phenotyping data were provided to us by the trial investigators in raw cell counts. The cell surface markers used to identify each subset is provided in Supplementary Data 8.

**Ex vivo stimulation and in vitro HIV-1 infectability assays**. Cryopreserved or fresh isolated PBMC samples from healthy donors ($n = 10$) were thaw and enriched for CD4 memory T cells by negative selection, according to manufacturer's protocol, with the EasySep Human CD4 memory T cell Enrichment Kit (StemCell Technologies, 10157). Isolated CD4 memory T cells were cultured at concentration of $2 \times 10^6$ cells/mL with logarithmic increasing concentrations of IFNα [0.02, 200 ng/mL], IFNβ [0.002, 200 ng/mL], IFNγ [0.005, 50 ng/mL] or left unstimulated. The efficacy of the cytokines to induce IRF7 and STAT1 phosphorylation was evaluated in 100,000 cells from 5 donors at the lowest and the highest concentration of the respective cytokines by flow cytometry. The remaining cells were kept in culture for 18 h. Following incubation, HIV-1 infection through spinoculation with 89.6 viral supernatant (NIH AIDS Reagent Program, Division of AIDS, NIAID, NIH: p89.6 from Ronald G. Collman, MD) was performed at 200 ng/mL p24/million CD4$^+$ T cells in the presence of 4 μg/mL polybrene (Sigma, H9268), at 2500 rpm for approximately 2.5 h at 30 °C. After spinoculation, viral supernatants were removed and infected cells were cultured at a concentration of $2 \times 10^6$ cells/mL in cRPMI supplemented with 30 IU/mL IL-2 (R&D Systems, 202-IL), 5 μM saquinavir (NIH Aids Reagent Program, 4658) and the respective cytokines at 37 °C, 5% $CO_2$ for three days. On day 4, HIV-1 p24 levels were evaluated by FCM on CD4-negative cells. The data are represented as the frequency of infected p24$^+$ cells, HIV-1 per cell level (median of fluorescence intensity) per condition

normalized by unstimulated cells. A paired Wilcoxon-rank sum was used to compared the frequencies/intensities of cells after interferon stimulation to the unstimulated condition.

**Integrative analysis**. A projection-based approach implemented in the R package mixOmics was adopted to assess the correlations between gene-expression and other data types (ICS, Luminex and FCM). For each pair of data type, a sparse-least square regression was used to project the first element of the pair onto the second element of the pair. Once the two data types are projected on the same scale, the Pearson correlation between the features of the two data types was calculated. To assess the probability of obtaining a Pearson correlation equal to or greater than the one observed, we derived a $p$ value based on the distribution of the Pearson correlations calculated between all pair of features of the two data types (i.e., the statistical universe). Pearson correlations corresponding to a $p$ value cutoff of 0.05 were considered significant.

**Deconvolution of blood transcriptome**. The function lsfit of the R package CellMix[19] was used to model the PBMC gene expression measures across samples as the function of contributions of immune subset-specific gene expression weighted by the corresponding cell frequencies of those subsets measured by FCM. To verify that the deconvolution was successful, we investigated the gene expression of the cell surface markers used for cell-sorting. We observed a concordant expression of several protein markers specifically expressed at the surface of those subsets (Supplementary Fig. 5a).

To obtain immune subset-specific gene expression estimates for RV144 controls and cases, we applied linear regression separately to each group's gene expression. The linear regression coefficient estimates are taken as surrogates for estimated cell type-specific average gene expression while a difference between these cell type-specific estimates was used as the level of gene expression change between the controls and cases in a given cell type.

**HIV-1 acquisition classifiers**. Logistic regression models were built using the function glm of the R package stats. Gender and behavior risk were included in all regressions as independent variables. HIV-1 acquisition, the dependent variable was coded as a binary variable while PFS, gene-expression pathways, IgA and IgG titers were coded as continuous variables. A Wald test was used to test if the coefficients of regression are statistically different zero (i.e., null hypothesis being no association between a marker of interest and HIV-1 acquisition). The accuracy of each regression model was estimated by tenfold cross-validation. Receiver operating characteristic curves and Delong's test was used to compare the accuracy of two regression models.

**Statistical analyses**. Student $t$ test was used to test for the significance of pathway and antibody response while a non-parametric test, the Wilcoxon rank-sum test, was used to test for difference in pathway expression between HIV-1 cases and controls. The Benjamini and Hochberg correction was used to adjust for multiple testing. for all gene expression analysis. A 5% cutoff on the probability of false positive (i.e., $p$ value) was used as a statistical stringency for all analyses presented in this work.

**Reporting summary**. Further information on experimental design is available in the Nature Research Reporting Summary linked to this article.

**Code availability**. All the source code used to generate the figures part of this manuscript is available at "https://github.com/sekalylab/rv144". The authors declare that all other data supporting the findings of this study are available from the authors upon request.

## Data availability

The microarray data have been submitted to the National Center for Biotechnology Information Gene Expression Omnibus ("https://www.ncbi.nlm.nih.gov/geo [https://www.ncbi.nlm.nih.gov/geo]") under accession number GEO: "GSE103740". The authors declare that all other data supporting the findings of this study are available from the authors upon request.

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

## Acknowledgements

We would like to thanks Petra Stafova for performing the gene-expression arrays experiments; Nicole Frahm and Stephen De Rosa for providing us with the intracellular and Luminex data; Barton Haynes for comments on the protocol and the paper. This work made use of the High-Performance Computing Resource in the Core Facility for Advanced Research Computing at Case Western Reserve University. Grants from the Bill and Melinda Gates Foundation (OPP1032325 and OPP1147555) supported this work. S.F. received a travel fellowship from the Bill and Melinda Gates Foundation (OPP1084285).

## Author contributions

J.K., P.P., S.N., S.R.-N., J.H.K., N.L.M. and M.J.M. were involved in the conceptualization and oversight of the study; R.T. and G.D.T provided the experimental data, S.F. was involved in conception of the methodology; S.F. was involved in software development; S.F., A.T. and F.L. were involved in formal analysis; F.B.T.P.L. and S.P.R. performed the ex vivo experiments; S.F., R.G. and R.-P.S. prepared the original draft; R.A.K., M.C., P.B.G., N.L.M. provided critical inputs.

## Additional information

**Competing interests:** The authors declare no competing interests.

