## [Peer Review File · Nature Communications]

Reviewers' Comments:

Reviewer #1:

Remarks to the Author:

Fourati et. all use a systems approach to analyze transcriptomic signatures of in vitro HIV-Env. stimulated PBMCs from RV144 vaccinated and placebo individuals. Understanding the mechanism of action of this vaccine is important. The problem is that the authors try to cover a lot of ground and although they describe some transcriptomic correlates of vaccination, these do not seem to improve response prediction, and in the light of previously published work on vaccine correlates the impact of these findings is unclear. One is unable to come away with a message or metric different from what we already know about this vaccine's efficacy. Another issue is that the paper as written is quite confusing- the flow is quite disjointed and hard to get through – there is a need to re-write the paper in a more logical, coherent and focused fashion.

The authors identify a signature of 4 major pathways associated with vaccination by employing linear regression models and performing gene set enrichment analysis. This analysis was limited in that it did not inform about the efficacy of the vaccine, so they go on to acquire an extended cohort including vaccinated individuals who are known to go on to develop HIV and perform linear regression models and GSEA on these individuals. One conclusion from this step is that the induction of ISGs is unique to those who don't later go on to contract HIV, a phenomenon not seen in the placebo samples indicating it may be vaccination specific. While there is a signature, I believe that the number of placebo individuals who go on to contract HIV (17 based on Supplementary table 1d) is too low a sample size to make this claim. The authors need to state these numbers upfront.

Later they use the mixOmics R package to identify correlations between transcriptomic, antibody response, and cytokine expression following immunization with the RV144 vaccine. However they do not state the correlation values and judging by the color, most seem to be low (<.5). This is a bit concerning since the strength of those correlations is important to make such claims.

They also performed a deconvolution analysis to check for what cell types were responsible for the expression of NFkB target genes. This analysis is not well documented, and nothing is cited to provide insight into the method. They appear to suggest that that correlation with cell frequency by FCM implies cell specific expression and this is a fundamentally flawed argument. In addition the methods section for the deconvolution cites Supplementary Fig. 4a which appears to be the wrong figure.

They then build a logistic regression model to predict acquisition of HIV-1 among RV144 vaccinees. This model takes into account some covariates and is tested by cross validation. In this model addition of IFNG pathway gene expression did not improve results which they claim is due to confounding effects, specifically the DPB1*13 allele. Stratification around this allele (Fig. 4B) however, only modestly improves results and although the correlation between IFNG response and IgG binding V1/V2 reaches a p value of <.05 the correlation coefficient of .34 is very poor and these results are hard to interpret.

Overall the question of gaining mechanistic insights into how the RV144 vaccine works is important, however it seems that the authors are drawing very loose correlative connections to make their point, and I'm not convinced about the validity/reproducibility of these claims.

Minor comments:

Line 129: "Interestingly, we also observed enrichment of putative binding sites for IRF7 in the promoter regions of genes associated with a reduced risk of HIV-1 acquisition (Supplementary Table 3c, 12/53 of ISG genes have a putative IRF7 binding site within \pm 2000 base pairs of their

transcriptional starting sites)....." I think the wrong figure is cited. Where is the information on binding sites shown?

Supplementary table 1: this should be simplified into one table. Distributing this information across four tables i.e. a-d is repetitive, and also confusing and distracting for the reader.

Line 868: Supplementary table 3b: "List of leading edges of the of 16/17 genesets significantly associated with HIV acquisition in vaccinees. Genesets in red are associated with vaccine-conferred protection while genesets blue are associated with HIV acquisition". Again the wrong table seems to be cited or the legend is wrong. There is no red/blue coloring in the cited table.

Reviewer #2:

Remarks to the Author:

The paper by Fourati et al describes an unbiased system biology approach used to define and understand mechanisms underlying vaccine mediated protection in the RV144 trial. Systems biology is performed from in vitro Env peptide stimulate cells collected before vaccination and 2 weeks after the last vaccination. Interestingly, this analysis shows clear distinction among major pathways that could be associated with reduced risk of infection (IFN γ pathway: genes associated with MHC class II maturation and antigen-presentation; and IRF7 pathway associated with innate antiviral program) and increased risk of infection (NF- κ B, mTORC1). Most importantly, integrative analysis including system biology data and previously reported association with infection risk in RV144 and showed a correlation of IFN γ pathway and CD4 ICS score and IgG to V1V2. On the other hand, the IFN γ pathway activation did not act as independent predictor of infection but plays a confounding role with vaccine induced cellular and humoral responses and the genetics (DPB1*13).

This work provides a very important novel contribution to our understanding of the complexity in deciphering correlates of protection in HIV.

Figure 2: please remove legend from within Figure and place outside
Lane 462: should be paired Wilcoxon-rank sum

Reviewer #3:

Remarks to the Author:

In this manuscript, Fourati et al describe the transcriptional responses in PBMCs ex vivo stimulated with Env peptide from individuals vaccinated with RV144. Overall this is a very interesting dataset that warrants detailed analysis and description. The transcriptional response reveals an association of IFN-related gene expression with protection of HIV infection in vaccinees. The authors further describe a set of analyses integrating transcriptional responses with previously measured immune parameters including polyfunctional TCs and serological responses. Further more in vitro infection studies are performed to validate the finding of a role of IFN α and IFN β induced IRF7-p, as well as to show an association of CD4 TCs stimulated with type I IFNs and the reduction of p24 positivity.

Major concerns

1) Overall, this is a very nice dataset, which provides already some interesting clues into the response of PBMCs from RV144 vaccinees. The manuscript is however extremely hard to follow. The transcriptional analysis design is insufficiently explained nor is it made clear in the results that this data is all based on an in vitro infection of PBMCs. This is a complicated setup and thus needs major

revisions. For example, the authors should include a clear and concise figure in the beginning outline what samples were taken and what groups are in this study. Moreover, the analysis steps need to be explained more clearly. Deciphering this manuscript is hard work and will be difficult for most readers not familiar with either HIV or transcriptomics.

2) I believe the transcriptional data deserves a bit more basic description as it is difficult to comprehend for readers not familiar with transcriptional data. This some visualization of the magnitude of the response in the pilot samples and remaining samples, and vaccine and placebo group and the corresponding HIV+ and HIV- participants. How many genes were differentially expressed?

3) The claims made by the authors are only partially supported by the results. Thus, if the authors for example claim that the data 'highlight the contribution of innate antiviral and cellular (NK/ADCC) functions to the correlates of risk results in RV144' (line 219) which the authors did not explicitly show. If they wish to conclude this they need to perform an ADCC assay using sera from the study. Similarly the authors claim to have found a IRF7-mediated mechanism '...whereby CD4+ T cells from subjects immunized with the RV144 vaccine will mount an Env specific type II interferon (interferon γ) response...'. This is incorrect. While the data suggests this, the authors should use IRF7 KO T cells and repeat the in vitro infection experiments.

Overall, it I suggest the authors spend more effort on the narrative including(!) the grammar (there are many mistakes!!!) in order to enhance the flow of the manuscript and make it clearer what was actually done. Additionally a list of minor concerns is below.

Minor concerns:

1) Abbreviations need to be spelled out first time they are used. E.g. HIV or PBMCs.

2) Line 69: Here 269 samples whereas the abstract states 223 samples.

3) Grammar! Consistency with expressions (i.e. in vitro vs in-vitro) and italic. Overall the grammar is poor in some sections and the sentences are sometimes extremely long (e.g. line 149+).

4) Line 77 sentence: As mentioned above this manuscript would benefit from a detailed and clear figure giving an overview of the samples and trial including a summary of how many samples in total, which belonged to the pilot which not, how many HIV neg vs. HIV pos in placebo and vaccine arm and time points.

5) Line 80: Of the 40 participants, what's the split in these between cases and non-cases?

6) Line 84 onwards. The authors talk about GSEA here and gene expression without making it clear that this is following in vitro stimulation and of which cells? Furthermore, the GSEA is not sufficiently explained as in what is the gene set database this was performed against, and was the gene list ranked by fold-change or significance levels?

7) Looking at the figure legends of Fig 1b it represents SLEA NOT GSEA. This is not made clear in the text (line 86)!

8) Fig 1 legend: It's not a histogram, but a bar graph. Why do the authors explain what a negatively enriched pathway is if none of them are negatively enriched? Are there negatively enriched pathways? IF so, they should be displayed. In this legend I see for the first time that this is a sample based enrichment in 1b – why is this not in the text? What is SLEA anyways, this is not sufficiently explained in the results text. Finally, 1c does not add up with the numbers stated in text (n). Authors write its 40 vaccinees and 10 placebo.....there are more than 40 and more than 10 placebo samples in the figure and its not explained what pre and post means!

9) Negative NES: How is this possible when ranked by p-value. Does this mean that these pathways are enriched by non-significantly induced genes – i.e. by background expression?

10) Line 89: The sentence 'genes downstream of mTORC1 as well as genes associated to allograft rejection i.e. genes triggered by T cell activation' makes little sense to me. Needs clarification and

some references.

- 11) Line 94 and 95. Please reference the context of the genes stated in brackets.
- 12) Line 111 – of the 30 placebo participants how many were HIV+ and HIV-?
- 13) Fig2 Legend: Need n in groups for placebo and vaccinees.
- 14) Line 114: 'Our analysis revealed that 3 out of the 4 pathways...'. In the figure legend its described as only 2 associated with acquisition?
- 15) Line 117: The authors start talking about ISGs but it is called Interferon_gamma_response in the figure. Please be consistent as this confuses the reader even more in an already complex study.
- 16) In addition to Fig 2, it would be helpful to plot the expression of the genes of the ISG response as this would give the reader a feeling of the expression levels and behavior of these genes. They may not all be significantly expressed, however it would make this more transparent.
- 17) Sentence in line 124: Needs references for context.
- 18) Line 127: "activation of the IRF7 innate antiviral program (IFIH1, IRF7)..". can the authors speak of a program if this consists only of 2 genes?
- 19) Line 128 to end of para: How was the TF analysis done? Could use a bit more explanation.
- 20) Line 143: 'These results suggest that induction of ISGs is a vaccine-induced correlate of protection from HIV-1...'. This is NOT a correlate of protection! It a) does not represent the mechanism of protection so would most likely only be surrogate of protection, and has not been validated. Please tone down the language!
- 21) Methodology is very cryptic and as mentioned before hard to understand. What for example is a 'Projection-based approach' (line 154).
- 22) Line 162: The authors only really report ONE pathway, and not pathways. What by the way about the other pathways – i.e. mTORC and NFkB etc.?
- 23) Line 167: How was the deconvolution performed?
- 24) Lone 172: This is the first time of mentioning Fig3 however none of the results are described here. This makes no sense – please revise.
- 25) Line 186: 'Only the PFS, the interaction between IgA:DQB1*06, the interaction between IgG:DPB1*13 and mTORC1 signaling remained statistically significantly associated to HIV-1 acquisition in a multivariate model.' Somewhat is odd in this sentence, do the authors mean 'interaction between IgA AND DQB1, or the interaction between IgA:DQB1 and mTORC1 signaling?
- 26) Line 192: Have the authors tried to only use the IFNg pathway alone to segregate the groups? The analysis without all the other correlates might be informative.
- 27) Supplemental Figure 6: No panel labels hence the legend is not informative.
- 28) Line 193: Not correct English.
- 29) Figure 4b,d and Figure 5b: Replacing the x-axis labels with HIV+ and HIV- would be useful.
- 30) Line 206: Please put reference to ADCC to put this in context with literature.
- 31) Line 215: Acquisition of what? Something missing here...
- 32) Line 215: Please explain in detail how this experiment was performed and what the markers mean etc. This is incomprehensible to a non HIV person!
- 33) Line 229 and Figure 5b: y-acis says TNFa signaling not NFkB signaling???
- 34) Fig 5b/d: Can the authors comment that some of the high responders and them possibly skewing these results?
- 35) Line 240 – last sentence of the Results section: "These results highlight the importance of controlling the balance between pro- and anti-inflammatory pathways to trigger the development of protective vaccine responses. " The authors have not shown this – please tone down.
- 36) Line 268: The sentence starting 'The IRF7 signature was enriched in IFNa-stimulated...' could go into results.
- 37) Line 312: '...confirms the role...' Please tone down –the data does not confirm this.
- 38) Paragraph Line 311: This paragraph seems random and utterly incomplete with little logic to the argumentation. XBP1 is an ER stress response pathway. How does ER stress factors support mTORC as a correlate of susceptibility? The mTOR pathway is an incredibly complex signaling pathway and its

primarily involved in sensing nutrients. So I do not understand how the authors make this connection without referencing anything. This seems a wildly speculative.

Reviewer # 1

Fourati et. all use a systems approach to analyze transcriptomic signatures of in vitro HIV Env. stimulated PBMCs from RV144 vaccinated and placebo individuals. Understanding the mechanism of action of this vaccine is important. The problem is that the authors try to cover a lot of ground and although they describe some transcriptomic correlates of vaccination, these do not seem to improve response prediction, and in the light of previously published work on vaccine correlates the impact of these findings is unclear.

We agree with reviewer #1 that the addition of transcriptomic pathways identified in our study as correlates of risk (IFN γ , NF- κ B and mTORC1 pathways) did not significantly improve prediction of HIV-1 acquisition among vaccinees. This being said, our study identified IRF7 as the upstream mediator of the innate immune response triggered by the RV144 vaccine. A gene expression signature that included IRF7 and its targets, including antiviral genes that can inhibit several steps of the HIV life cycle, was associated with lower risk of HIV-1 acquisition among RV144 vaccinees. We confirmed experimentally in an in vitro system of HIV-1 infection of primary T cells that IRF7 is associated with a reduced rate of infection which could lead to a reduced risk of HIV-1 acquisition. The implication of IRF7 in the mechanism of the response to RV144 is a novel and important finding that was deduced from the transcriptomic analysis. We have also shown that a distinct set of genes regulated by STAT1, downstream of type II interferons, correlates with PFS, another correlate of protection of HIV-1 acquisition.

In contrast to IRF7, the mTORC1 pathway remained significantly associated with HIV-1 acquisition in a multivariate model that included previously reported correlates of risk of HIV-1 acquisition, suggesting that mTORC1 provides independent information that was not previously reported. The implication of mTORC1 pathway in the development of poor responses to vaccines has been reported¹ as rapamycin an inhibitor of mTORC1 restored in vivo the response to Flu vaccination in a cohort of elderly subjects. Our findings open the venue for novel interventions such as low dose rapamycin that can improve the response to RV144 and future HIV vaccines.

One is unable to come away with a message or metric different from what we already know about this vaccine's efficacy.

We now highlight, in the abstract and in the results section, the mechanisms underlying lower and higher risk of HIV-1 acquisition among RV144 vaccinees as important original findings that pave the way for novel interventions that can improve vaccine efficacy. See response to the previous comment.

Another issue is that the paper as written is quite confusing, the flow is quite disjointed and hard to get through – there is a need to rewrite the paper in a more logical, coherent and focused fashion.

We agree with reviewer #1 and rewrote the results section to be more fluid. We also provided as  a workflow that describes all the steps of our bioinformatic analysis.

The authors identify a signature of 4 major pathways associated with vaccination by employing linear regression models and performing gene set enrichment analysis. This analysis was limited in that it did not inform about the efficacy of the vaccine, so they go on to acquire an extended cohort including vaccinated individuals who are known to go on to develop HIV and perform

linear regression models and GSEA on these individuals. One conclusion from this step is that the induction of ISGs is unique to those who don't later go on to contract HIV, a phenomenon not seen in the placebo samples indicating it may be vaccination specific. While there is a signature, I believe that the number of placebo individuals who go on to contract HIV (17 based on Supplementary table 1d) is too low a sample size to make this claim. The authors need to state these numbers upfront.

The number of placebos used is 30 participants. While an n=30 participants is not large, it is still of a considerable size for a transcriptomics study. This being said, we performed a bootstrapping approach where we selected 17 vaccinees that acquired HIV and 13 vaccinees that remained HIV negative and tested whether we could find association with HIV acquisition with a total number of participants of 30 (same number as for the placebo recipients). We performed this exercise 100 times and in 64 out of the 100 iterations, we were able to detect significant association of the gene-expression signature with the risk of HIV-1 acquisition. This suggests that the lower number of placebos is unlikely to be the sole explanation for the lack of association between IFN γ response genes and HIV acquisition. We have modified the text to state the number of placebo recipients that acquired HIV-1 and those that remained HIV-1 negative during the follow-up in the results section of the manuscript.

Later they use the mixOmics R package to identify correlations between transcriptomic, antibody response, and cytokine expression following immunization with the RV144 vaccine. However, they do not state the correlation values and judging by the color, most seem to be low (<.5). This is a bit concerning since the strength of those correlations is important to make such claims.

We provide the correlation (and corresponding p-values) between the markers as labels of the network in Supplementary Fig. 4. Although some correlations showed a r value below 0.5 (IFN γ response and IgG against V1/V2: $r=0.218$ and IFN γ response to Env-specific CD4+ T cells: $r=0.286$) these were highly significant ($p \in [0.01 \text{ to } 0.00333]$).

They also performed a deconvolution analysis to check for what cell types were responsible for the expression of NF κ B target genes. This analysis is not well documented, and nothing is cited to provide insight into the method. They appear to suggest that that correlation with cell frequency by FCM implies cell specific expression and this is a fundamentally flawed argument.

The details of the strategy used for the deconvolution are provided in the material and method section and cited the corresponding reference in the result section. The `lsfit` function of the R package CellMix² was used to model the PBMC gene expression measures across samples as a function of contributions of immune subset-specific gene expression weighted by the corresponding cell frequencies measured by flow cytometry. This strategy has been used and published in numerous manuscripts by our group and others^{3,4}. We performed the deconvolution analysis to demonstrate, as requested by the reviewer, the association of monocytes with the NF- κ b gene expression signature; we did not limit ourselves to correlating gene expression with cell frequencies.

In addition the methods section for the deconvolution cites Supplementary Fig. 4a which appears to be the wrong figure.

This was corrected in the resubmitted version of the manuscript (Supplementary Fig. 5a).

They then build a logistic regression model to predict acquisition of HIV1 among RV144 vaccinees. This model takes into account some covariates and is tested by cross validation. In this model addition of IFNG pathway gene expression did not improve results which they claim is due to confounding effects, specifically the DPB1*13 allele. Stratification around this allele (Fig. 4B) however, only modestly improves results and although the correlation between IFNG response and IgG binding V1/V2 reaches a p value of <.05 the correlation coefficient of .34 is very poor and these results are hard to interpret.

As pointed by reviewer # 1, we show that the IFN γ pathway is not an independent correlate of risk when adjusting for previously reported markers. We agree with reviewer # 1 that the correlation between IFN γ and IgG among DPB1*13 is moderate but statistically significant (Pearson correlation $r=0.340$ $p=0.0119$); we think this may cause the lack of significance of IFN γ in the multivariate model. To further support this point we build a stepwise multivariate model with IFN γ alone, IgG:DBP13 alone and IFN γ + IgG:DBP13 and show that the IFN γ pathway loses its significance when IgG:DBP13 is present in the model (Supplementary Table 3). This new analysis further highlights that IFN γ and IgG:DPB1*13 are correlated to each other; it suggests that IFN γ might have increased antigen presentation of a specific T cell epitope restricted by DPB1 which in turn could have resulted in a higher IgG response to HIV.

Overall the question of gaining mechanistic insights into how the RV144 vaccine works is important, however it seems that the authors are drawing very loose correlative connections to make their point, and I'm not convinced about the validity/reproducibility of these claims.

Haynes et al.⁵ and Lin et al.⁶ used the same dataset that ours. That sample size was obtained after thorough power analysis. This dataset size (263 participants) is one the biggest human transcriptomic vaccine response dataset published so far. All our transcriptomic results were confirmed using other experimental approaches (Flow cytometry, ICS, Luminex...).

We also highlighted in the abstract and in the result section the experimental validation that confirmed the findings of the transcriptomic analysis (namely IRF7). We show that both type I and type II interferons upregulate expression of pIRF7 and this is associated to an 8 fold resistance to HIV-1 infection. Finally we want to point out that the data and computer code used to generate our analysis have been made public to show that our findings can be reproduced by other groups.

Minor comments:

Line 129: “Interestingly, we also observed enrichment of putative binding sites for IRF7 in the promoter regions of genes associated with a reduced risk of HIV1 acquisition (Supplementary Table 3c, 12/53 of ISG genes have a putative IRF7 binding site within ± 2000 base pairs of their transcriptional starting sites).....” I think the wrong figure is cited. Where is the information on binding sites shown?

The correct figure was cited: “We also observed... (Supplementary Table 3c, 12/53 of IFN γ response genes are also part of the geneset V\$IRF7_01 having a putative IRF7 binding site within ± 2000 base pairs of their transcriptional starting sites)... and genes induced in IRF7 overexpression experiments (Supplementary Table 3c). “. This statement refers to the MSigDB geneset V\$IRF7_01. We clarified the statement in the text.

Supplementary table 1: this should be simplified into one table. Distributing this information across four tables i.e. ad is repetitive, and also confusing and distracting for the reader.

We now provide the number of participants included in each dataset in Fig. 1. We kept the information presented in Supplementary Table 1 to describe the characteristics of the cohorts when compared to the parent RV144 cohort and after stratification by treatment (vaccine or placebo). We believe this information is necessary to understand that transcriptomic pilot and case/control datasets share similar clinical characteristics with the parent RV144 cohort.

Line 868: Supplementary table 3b: “List of leading edges of the of 16/17 genesets significantly associated with HIV acquisition in vaccinees. Genesets in red are associated with vaccine conferred protection while genesets blue are associated with HIV acquisition”. Again the wrong table seems to be cited or the legend is wrong. There is no red/blue coloring in the cited table. This was corrected in the resubmitted version of the manuscript (Supplementary Table 3).

Reviewer #2 (Remarks to the Author):

Figure 2: please remove legend from within Figure and place outside
We agree with reviewer #2 and placed the legend outside of Fig. 3 (former Figure 2).

Lane 462: should be paired Wilcoxon rank-sum
This was corrected in the resubmitted version of the manuscript.

Reviewer #3 (Remarks to the Author):

Major concerns

1) Overall, this is a very nice dataset, which provides already some interesting clues into the response of PBMCs from RV144 vaccinees. The manuscript is however extremely hard to follow. The transcriptional analysis design is insufficiently explained nor is it made clear in the results that this data is all based on an *in vitro* infection of PBMCs. This is a complicated setup and thus needs major revisions. For example, the authors should include a clear and concise figure in the beginning outline what samples were taken and what groups are in this study. Moreover, the analysis steps need to be explained more clearly. Deciphering this manuscript is hard work and will be difficult for most readers not familiar with either HIV or transcriptomics.

We are pleased that reviewer #3 qualified data presented herein as providing interesting clues into response to the RV144 vaccine. We agree with reviewer #3 and now clearly state in the result section that the transcriptomic analysis was performed using PBMC stimulated with HIV-1 Env peptides: “We compared the transcriptomic profile of *in vitro* HIV-1 Env-stimulated PBMCs... (Supplementary Table 1a-b)”. We also have provided further description of the methods and also have included as Fig. 1, a workflow illustrating the steps of our analysis.

Fig. 1. Study overview. Four steps were used in a bioinformatic strategy aimed at identifying transcriptomic markers of risk of HIV-1 acquisition among RV144 vaccinees. A first transcriptomic dataset of PBMCs collected from 40 vaccinees and 10 placebo recipients pre-vaccination and two weeks after vaccination and stimulated with Env peptides was used to identify pathways modulated by the RV144 vaccine (step 1). A second independent transcriptomic dataset of blood collected from 183 vaccinees and 30 placebo, two weeks after vaccination was used to identify pathways associated with HIV-1 acquisition. Logistic regression was used to build multi-OMICS classifier of HIV-1 acquisition among RV144 vaccinees (step 3) and a project-based integrative analysis was used to associated those different OMICS to identify mechanistic mediator of vaccine response (step 4).

2) I believe the transcriptional data deserves a bit more basic description as it is difficult to comprehend for readers not familiar with transcriptional data. This some visualization of the magnitude of the response in the pilot samples and remaining samples, and vaccine and placebo group and the corresponding HIV+ and HIV participants. How many genes were differentially expressed?

We agree with reviewer #3 and added to the text (results section) the number of genes differentially expressed after vaccination in the results section. 2946 genes were altered post-immunization and differentially expressed between the vaccine and placebo groups (LIMMA: moderated t -test $p \leq 0.05$). 2058 and 3009 genes were differentially expressed between the cases and controls of the placebo and vaccine groups respectively (LIMMA: t -test $p \leq 0.05$).

3) The claims made by the authors are only partially supported by the results. Thus, if the authors for example claim that the data ‘highlight the contribution of innate antiviral and cellular (NK/ADCC) functions to the correlates of risk results in RV144’ (line 219) which the authors did not explicitly show. If they wish to conclude this they need to perform an ADCC assay using sera from the study.

The association of ADCC with reduced risk of HIV acquisition was previously published in Haynes et al 2012. Samples from the same participants used in Haynes et al.⁵ were used in our current study. A subsequent study proved the involvement of ADCC in RV144 vaccine efficacy⁷. This point and the corresponding reference are now clearly stated in the text.

Similarly the authors claim to have found a IRF7 mediated mechanism ‘...whereby CD4+ T cells from subjects immunized with the RV144 vaccine will mount an Env specific type II interferon (interferon γ) response...’. This is incorrect. While the data suggests this, the authors should use IRF7 KO T cells and repeat the in vitro infection experiments.

While we agree with reviewer # 3 that KO experiments is the ultimate and definite strategy to show causation, we show in dose response experiments that the resistance to HIV-1 infection conferred by type I and type II interferons is dose dependent which provides further evidence to support the proposed mechanisms. In addition resistance to infection conferred by type I and type II interferons is indicated by the upregulation of several genes well known to trigger resistance to HIV-1 infection⁸⁻¹². We have modified the text to iterate that our experimental validation supports our transcriptomic findings, but do not prove that IRF7 is required for vaccine mediated protection. In a subsequent and separate study we plan to test the knock-out of IRF7 from RV305 cells (follow up to the RV144 study) which could potentially make those T cells more susceptible to HIV-1 acquisition.

Overall, it I suggest the authors spend more effort on the narrative including(!) the grammar (there are many mistakes!!!) in order to enhance the flow of the manuscript and make it clearer what was actually done.

We agree with reviewer #3 and rewrote the results section to be clearer.

Additionally a list of minor concerns is below.

Minor concerns:

1) Abbreviations need to be spelled out first time they are used. E.g. HIV or PBMCs.

This was corrected in the resubmitted version of the manuscript.

2) Line 69: Here 269 samples whereas the abstract states 223 samples.

Line 69 states 263 which is the sum of the 223 vaccinees and 40 placebos listed in the abstract. We added the separation at line 69 to be clearer.

3) Grammar! Consistency with expressions (i.e. in vitro vs invitro)

and italic. Overall the grammar is poor in some sections and the sentences are sometimes extremely long (e.g. line 149+).

We uniformized the use of expressions throughout the manuscript, and split the sentence at line 149.

4) Line 77 sentence: As mentioned above this manuscript would benefit from a detailed and clear figure giving an overview of the samples and trial including a summary of how many samples in total, which belonged to the pilot which not, how many HIV neg vs. HIV pos in placebo and vaccine arm and time points.

We agree with reviewer #3 and provided as , a workflow illustrating the steps of our analysis and samples used at each step.

5) Line 80: Of the 40 participants, what's the split in these between cases and non cases?

All 40 vaccinees included in the RV144 pilot transcriptomic study were HIV-1 negative at the last follow-up visit. This information is provided in  and was now added to the result section.

6) Line 84 onwards. The authors talk about GSEA here and gene expression without making it clear that this is following in vitro stimulation and of which cells? Furthermore, the GSEA is not sufficiently explained as in what is the gene set database this was performed against, and was the gene list ranked by fold change or significance levels?

The previous sentence (line 78) state that *in vitro* stimulated PBMC of 40 vaccinees and 10 placebo-recipients were compared. The significance cutoff used, i.e. $FDR \leq 5\%$, is stated at line 84. We added the database used for the GSEA analysis in the main result requested by reviewer #3. The metric used to rank genes was LIMMA t-statistic which is now more clearly stated in the material & methods section.

7) Looking at the figure legends of Fig 1b it represents SLEA NOT GSEA. This is not made clear in the text (line 86)!

A description of the SLEA method was added to the results and material & methods section.

8) Fig 1 legend : It's not a histogram, but a bar graph. Why do the authors explain what a negatively enriched pathway is if none of them are negatively enriched? Are there negatively enriched pathways?

With bar charts, each column represents a group defined by a categorical variable; and with histograms, each column represents a group defined by a continuous, quantitative variable. Fig 2a is indeed a bar plot and not a histogram. We agree with reviewer # 3 that no genesets showed negative enrichment. We removed the section explaining the significance of a negative NES .

IF so, they should be displayed. In this legend I see for the first time that this is a sample based enrichment in 1b – why is this not in the text? What is SLEA anyways, this is not sufficiently explained in the results text.

A description of the SLEA method was added to the results and material & methods section.

Finally, 1c does not add up with the numbers stated in text (n). Authors write its 40 vaccinees and 10 placebo.....there are more than 40 and more than 10 placebo samples in the figure and its not explained what pre and post means!

The heatmap shows the expression profile pre and post vaccination for the 40 vaccinees and 10 placebo recipients (a total of 100 columns). The legend of Fig 2c was rewritten to state those numbers.

9) Negative NES: How is this possible when ranked by p-value. Does this mean that these pathways are enriched by non significantly induced genes – i.e. by background expression?

A description of the GSEA method and NES score was added to the results and material & methods section and the use of LIMMA t-statistic for pre-ranking genes is now clearly stated in the material & methods section. A negative NES corresponds to an enrichment among genes repressed post-vax when compared to pre-vax.

10) Line 89: The sentence ‘genes downstream of mTORC1 as well as genes associated to allograft rejection i.e. genes triggered by T cell activation’ makes little sense to me. Needs clarification and some references.

We agree with reviewer # 3 and provided references in Supplementary Table 2e.

11) Line 94 and 95. Please reference the context of the genes stated in brackets.
The references in Supplementary Table 2 are now placed in the text of the results section.

12) Line 111 – of the 30 placebo participants how many were HIV+ and HIV?
13 placebos remained HIV negative while 17 acquired HIV. The information was provided in Supplementary Table 1 and now is also added to the results section.

13) Fig2 Legend: Need n in groups for placebo and vaccinees.
We agree with reviewer # 3 and added the number of participants per group in the Fig. 3 (former Fig. 2) legend.

14) Line 114: ‘Our analysis revealed that 3 out of the 4 pathways...’. In the figure legend its described as only 2 associated with acquisition?
We clarified that among the 3 associated with HIV acquisition, 2 were associated with an increased risk, while 1 was associated with a lower risk.

15) Line 117: The authors start talking about ISGs but it is called Interferon_gamma_response in the figure. Please be consistent as this confuses the reader even more in an already complex study.
We defined in the first paragraph of the result section ISGs as “IFN γ stimulated genes”, but we agree with reviewer #3 and used IFN γ response pathway throughout the paper for clarity purposes.

16) In addition to Fig 2, it would be helpful to plot the expression of the genes of the ISG response as this would give the reader a feeling of the expression levels and behavior of these genes. They may not all be significantly expressed, however it would make this more transparent.
We agree with reviewer #3 and added a supplementary figure with the heatmap of IFN γ response (Supplementary Fig. 2).

17) Sentence in line 124: Needs references for context.
The references in Supplementary Table 3 are now in the text of the results section.

18) Line 127: “activation of the IRF7 innate antiviral program (IFIH1, IRF7).’ can the authors speak of a program if this consists only of 2 genes?
We provide two examples in the main text and more examples of IRF7 target genes in Supplementary Fig. 2. Supplementary Fig. 2 is now cited.

19) Line 128 to end of para: How was the T F analysis done? Could use a bit more explanation.
Genes with binding sites for TF were obtained from the C3 module of the MSigDB database. This is now stated at line 128.

20) Line 143: ‘These results suggest that induction of ISGs is a vaccine induced correlate of protection from HIV1...’. This is NOT a correlate of protection! It a) does not represent the

mechanism of protection so would most likely only be surrogate of protection, and has not been validated. Please tone down the language!

We now use the term “correlates of reduced risk of acquisition” at line 143.

21) Methodology is very cryptic and as mentioned before hard to understand. What for example is a ‘Projection-based approach’ (line 154).

A description of the mixOmics strategy is now given in the results and the material & methods section.

22) Line 162: The authors only really report ONE pathway, and not pathways. What by the way about the other pathways – i.e. mTORC and NFκB etc.?

The typo was corrected in a resubmitted version of the manuscript. We provided a description of IFN γ (the only pathway associated with reduced risk) and then in the subsequent paragraph we provided further details about mTORC and NF- κ b (associated with an increased risk).

23) Line 167: How was the deconvolution performed?

A brief description of the deconvolution analysis performed is provided in the results section, while a more detailed explanation is provided in the material & methods section. This strategy has been used and published in numerous manuscripts by our group and others^{3,4}.

24) Line 172: This is the first time of mentioning Fig3 however none of the results are described here. This makes no sense – please revise.

We described further **Fig. 4** (former Fig. 3) in the results section in the resubmitted version of the manuscript.

25) Line 186: ‘Only the PFS, the interaction between IgA:DQB1*06, the interaction between IgG:DPB1*13 and mTORC1 signaling remained statistically significantly associated to HIV1 Acquisition in a multivariate model.’ Somewhat is odd in this sentence, do the authors mean ‘interaction between IgA AND DQB1, or the interaction between IgA:DQB1 and mTORC1 signaling?’

In this sentence we replaced “interaction between” by “interaction term”. The interaction term IgG:DPB1*13 allow to assess the association between IgG level with HIV-1 acquisition separately for DPB1*13- and DPB1*13+ vaccinees. The same statistical analysis was used by Prentice et al that identified a better association of antibody response with HIV-1 acquisition when RV144 vaccinees are stratified by HLA haplotype¹³.

26) Line 192: Have the authors tried to only use the IFN γ pathway alone to segregate the groups? The analysis without all the other correlates might be informative.

We indeed performed this analysis and a predictive model with IFN γ (when adjusting for gender and behavior risk “BRA_risk”) but without the previously reported correlates of risk IgA:DQB1*06, IgG:DBP1*13 and PFS. That analysis shows that indeed IFN γ pathway is significantly associated with lower risk of HIV-1 acquisition among RV144 vaccinees (Odds ratio=0.883 and p=0.00837). This analysis is now shown as **Supplementary Table 5**. However, because previous correlates were reported, we must compare the IFN γ response to the best performing models reported so far to provide its clinical usefulness.

27) Supplemental Figure 6: No panel labels hence the legend is not informative.
Supplemental Fig. 6 was replaced with Supplementary Table 5.

28) Line 193: Not correct English.
The sentence at line 193 was rephrased.

29) Figure 4b,d and Figure 5b: Replacing the xaxis labels with HIV+ and HIV would be useful.
Fig. 4 and Fig. 5 (former Figs. 4 and 5) were modified accordingly.

30) Line 206: Please put reference to ADCC to put this in context with literature.
Genes implicated in ADCC were derived from the geneset KEGG_NATURAL_KILLER_CELL_MEDIATED_CYTOTOXICITY. Reference to Supplementary Table 5c was added to the main text.

31) Line 215: Acquisition of what? Something missing here...
“HIV” was added at line 215.

32) Line 215: Please explain in detail how this experiment was performed and what the markers mean etc. This is incomprehensible to a non HIV person!
A better detail of the experiment performed was added to the legend of Fig. 5d: The natural target of HIV-1 is CD4+ cells. Once CD4+ cells are infected they expressed HIV-1 protein p24 and the expression of CD4 at the cell surface is reduced (CD4⁻).

33) Line 229 and Figure 5b: y-axis says TNFa signaling not NFkB signaling???
The full name of the pathway is “TNFa signaling via NF- κ b”. To avoid confusion we used a uniform name throughout the paper.

34) Fig 5b/d: Can the authors comment that some of the high responders and them possibly skewing these results?
We agree with reviewer #3 that 2 out of 10 donors show heightened frequencies of infected T cells (CD4-p24+) when treated with increasing concentration of interferons. The hypothesis tested in these experiments is that interferon treatment will reduce the number of T cells infected by HIV-1, which is true for 8/10 donors. We did not identify any a priori characteristics that would lead us to believe that those two donors would behave as outliers. Removing those two donors with results opposite to our initial hypothesis would only improve the fold reduction of T cells infected by HIV-1 after interferon treatment.

35) Line 240 – last sentence of the Results section: “These results highlight the importance of controlling the balance between pro and anti-inflammatory pathways to trigger the development of protective vaccine responses. “ The authors have not shown this – please tone down.
We agree and the sentence was modified accordingly.

36) Line 268: The sentence starting ‘The IRF7 signature was enriched in IFN α stimulated...’ could go into results.
We agree and added that sentence to the result section.

37) Line 312 : ‘...confirms the role...’ Please tone down –the data does not confirm this.
We agree and the sentence was modified accordingly.

38) Paragraph Line 311: This paragraph seems random and utterly incomplete with little logic to the argumentation. XBP1 is an ER stress response pathway. How does ER stress factors support mTORC as a correlate of susceptibility? The mTOR pathway is an incredibly complex signaling pathway and its primarily involved in sensing nutrients. So I do not understand how the authors make this connection without referencing anything. This seems a wildly speculative.
We agree and the sentence was modified accordingly.

References

1. Mannick, J. B. *et al.* mTOR inhibition improves immune function in the elderly. *Sci Transl Med* **6**, 268ra179 (2014).
2. Gaujoux, R. & Seoighe, C. CellMix: A comprehensive toolbox for gene expression deconvolution. *Bioinformatics* **29**, 2211–2212 (2013).
3. Shen-Orr, S. S. *et al.* Cell type-specific gene expression differences in complex tissues. *Nat. Methods* **7**, 287–289 (2010).
4. Vaccari, M. *et al.* HIV vaccine candidate activation of hypoxia and the inflammasome in CD14+ monocytes is associated with a decreased risk of SIVmac251 acquisition. *Nat. Med.* (2018). doi:10.1038/s41591-018-0025-7
5. Haynes, B. F. *et al.* Immune-correlates analysis of an HIV-1 vaccine efficacy trial. *N Engl J Med* **366**, 1275–1286 (2012).
6. Lin, L. *et al.* COMPASS identifies T-cell subsets correlated with clinical outcomes. *Nat Biotechnol* **33**, 610–616 (2015).
7. Tomaras, G. D. *et al.* Vaccine-induced plasma IgA specific for the C1 region of the HIV-1 envelope blocks binding and effector function of IgG. *Proc Natl Acad Sci U S A* **110**, 9019–9024 (2013).
8. Guzzo, C., Fox, J. C., Miao, H., Volkman, B. F. & Lusso, P. Structural Determinants for the Selective Anti-HIV-1 Activity of the All-beta Alternative Conformer of XCL1. *J Virol* **89**, 9061–9067 (2015).
9. Yu, J. *et al.* IFITM Proteins Restrict HIV-1 Infection by Antagonizing the Envelope Glycoprotein. *Cell Rep* **13**, 145–156 (2015).
10. Durfee, L. A., Lyon, N., Seo, K. & Huijbregtse, J. M. The ISG15 conjugation system broadly targets newly synthesized proteins: implications for the antiviral function of ISG15. *Mol Cell* **38**, 722–732 (2010).
11. Dicks, M. D. *et al.* Oligomerization Requirements for MX2-Mediated Suppression of HIV-1 Infection. *J Virol* **90**, 22–32 (2015).
12. Uchil, P. D., Quinlan, B. D., Chan, W. T., Luna, J. M. & Mothes, W. TRIM E3 ligases interfere with early and late stages of the retroviral life cycle. *PLoS Pathog* **4**, e16 (2008).
13. Prentice, H. A. *et al.* HLA class II genes modulate vaccine-induced antibody responses to affect HIV-1 acquisition. *Sci Transl Med* **7**, 296ra112 (2015).

Reviewers' Comments:

Reviewer #1:

Remarks to the Author:

Fourati et. present a revised version of their manuscript. The previous draft had issues with the authors trying to cover a lot of ground and hinging key findings on small effect sizes. In addition, the structure of the paper was confusing, and thus it was difficult to come away with a message or metric different from what we already know about this vaccine's efficacy. In the revised version, the authors address some of my previous issues but many still remain. Attempts at revisions are inadequate and efforts at simplifying the manuscript for the reader are minimal. The paper is very hard to follow and incoherent requiring repeated readings to dissect what the authors are trying to convey and what their data mean. Conclusions are overstated in most parts and key claims are lacking mechanistic experimental evidence. Tables have subparts, but in many places the reader is directed to the whole table leaving it up to the reader to decode what the authors are actually referring to. Systems level transcriptomic findings are not distilled down to key impactful insights or conveyed in a comprehensible manner. The paper is still replete with sentence construction and grammatical errors. An extensive line by line, issue by issue coverage of these problems will run into pages therefore I am highlighting some of the major broad issues pertaining to why I believe the paper remains unsuitable for publication. I emphasize that I do believe that understanding the mechanism of action of this vaccine is important and recognize that changes have been made to the paper in response to the reviews (Fig. 1 especially clarifies the study design) but these are quite inadequate and the authors don't get us close to a convincingly supported set of conclusions. My overall view is that a thorough rewriting of the paper is needed to make this paper accessible to the reader and greater experimental evidence is needed to support central claims.

Major concerns:

1. The authors now focus in on identification of IRF7 as a mediator of the innate response to the RV144 vaccine. There is no convincing knockdown or KO evidence to support this claim. A similar experiment for mTORC1 would be helpful and is not attempted. The authors present dose response data showing that IRF7 is phosphorylated in response to IFNs and IFN treatment renders cells resistant to HIV infection. This is entirely expected and unsurprising and does not convincingly show that IRF7 and its target genes are executioners of this protection or even mechanistically important in the process.
2. In response to concerns over effect size and very low correlation values the authors argue that they are statistically significant without addressing the implications of such a small effect. For instance: In Supplementary Fig. 7b, $r=0.412$, therefore $r^2 = 0.169744$. In Supplementary Fig. 4, for IFN γ signaling response pathway correlation to Env-specific CD4+ T cells, $r=0.286$, therefore $r^2 = 0.081796$, thus only $\sim 8.2\%$ of variance is explained by the IFN γ response. In addition no differential expression results reach significance after multiple testing likely highlighting why GSEA and other metrics were used. This is a major concern about the biological meaning and validity of these findings and indicative of high false positives in the claims being made here.
3. The discussion is poorly written and several arguments are still difficult to understand in context and make no sense. It could be that they are just not phrased properly and getting lost in communication - the authors need to work on this. For example: line 353-"The presence of cholesterol as one of the pathways that is positively associated to acquisition...." - this sentence makes no sense in the flow of the discussion - it is unclear why the authors are talking about cholesterol which is not mentioned anywhere previously in the results and what this line means in context. The text has many such issues.
4. Supplemental tables for example Table S2, S3 or S6 have multiple parts but the relevant subparts are not always cited in the text making these difficult to know what panel the authors are referring

to when they refer to 'Supplementary Table 2 or 3'.

My opinion is that the results of this study need to be interpreted in context of what is already published in the field. Given that other viral transcriptomic vaccination studies have been able to separate out clearer more robust correlations of gene expression to AB response, cellularity etc. and do not rely on such loose correlative measures, these findings of this should be taken with precaution especially when lacking lots of mechanistic experimental evidence.

Reviewer #3:

Remarks to the Author:

The authors have reworked the requested bits and added information clarifying the study and analyses performed. It now reads a lot easier, and the analyses are a lot clearer. Below are a few final minor comments:

In the abstract the authors write that RV144 is the vaccine but in the first line of the intro they write that RV144 was a trial. This is confusing, please be consistent (was the trial called the same as the vaccine or what is happening here).

In the discussion the authors mentioned the ALVAC vector again which was only mentioned in the introduction. This seems a little random and might deserve a little bit more context.

Also the authors mentioned that in HIV+ participants the NFkB and mTORC pathway were associated with infection. Their wording that these pathways are required for HIV infection and hence these participants did not benefit from the vaccination is not coherent. I would suggest to reword this slightly to make sure the reader does not implicate causation.

There are several supplementary multipage PDFs which I cannot decipher. The file names are inconclusive, and it's a mix of tables and figures. Some tables are completely useless as they cannot be read, others seem more readable however I can't tell what they refer to. Please fix this.

Finally, while I am happy that the authors addressed all comments, I lack structure in doing so. The authors have obviously forgotten to indicate the changed text passages in the rebuttal letter and since the original page and line numbers have certainly changed due to additions to the text, this was hard work. Please make this easier in the future as this delays the review significantly!

Reviewer #1

1. The authors now focus in on identification of IRF7 as a mediator of the innate response to the RV144 vaccine. There is no convincing knockdown or KO evidence to support this claim. A similar experiment for mTORC1 would be helpful and is not attempted. The authors present dose response data showing that IRF7 is phosphorylated in response to IFNs and IFN treatment renders cells resistant to HIV infection. This is entirely expected and unsurprising and does not convincingly show that IRF7 and its target genes are executioners of this protection or even mechanistically important in the process.

A statement about the need for additional experiments to further understand the contribution of the IFN γ pathway in RV144 mediated protection and of the mTORC1 pathway in risk of HIV-1 acquisition has been added in the discussion section (lines 473 to 475).

2. In response to concerns over effect size and very low correlation values the authors argue that they are statistically significant without addressing the implications of such a small effect. For instance: In Supplementary Fig. 7b, $r=0.412$, therefore $r^2 = 0.169744$. In Supplementary Fig. 4, for IFN γ signaling response pathway correlation to Env-specific CD4+ T cells, $r=0.286$, therefore $r^2 = 0.081796$, thus only ~8.2 % of variance is explained by the IFN γ response. In addition no differential expression results reach significance after multiple testing likely highlighting why GSEA and other metrics were used. This is a major concern about the biological meaning and validity of these findings and indicative of high false positives in the claims being made here.

We agree with reviewer #1 about the small effect observed and for transparency, we provide both the Pearson correlations and associated p-values for each significant correlation between different OMICs in Fig. 5, Supplemental Fig. 4 and Supplemental Fig. 7. We added sentences in the discussion stipulating the amount of IFN γ response variance explained by previously identified correlates of the risk of HIV-1 acquisition in RV144 vaccinees (lines 355 to 357).

The absence of significant univariate gene association with HIV-1 acquisition among RV144 vaccinees is mentioned in the result section (line 165). Pathway level approaches such as GSEA have proven to be more statistically powerful to extract mechanistic insights for transcriptomic data than gene-level analysis (cf. Subramanian A. et al., 2015 #Comparing two studies of lung cancer). Thus it not surprising that we identify pathways associated with the risk of HIV-1 acquisition rather than single gene associations in our dataset. Also, the integrative analysis and our ex vivo experiments confirm some of the observations we made based on the pathway-level analysis.

We strongly agree with reviewer #1 that additional experiments need to be performed to fully ascertain the pathways identified in the current transcriptomic study and thus we added at the end of the paragraph discussing the limitations of our study that additional mechanistic validations are required (lines 473 to 475).

3. The discussion is poorly written and several arguments are still difficult to understand in context and make no sense. It could be that they are just not phrased properly and getting lost in communication - the authors need to work on this. For example: line 353-“The presence of cholesterol as one of the pathways that is positively associated to acquisition...” – this sentence makes no sense in the flow of the discussion - it is unclear why the authors are talking about

cholesterol which is not mentioned anywhere previously in the results and what this line means in context. The text has many such issues.

We agree with reviewer #1 and removed the paragraph about the potential mechanisms implicating cholesterol synthesis in risk of HIV-1 acquisition as the pathway was not highlighted in the results section of the paper and would require further experimental validation (lines 418 to 419).

4. Supplemental tables for example Table S2, S3 or S6 have multiple parts but the relevant subparts are not always cited in the text making these difficult to know what panel the authors are referring to when they refer to 'Supplementary Table 2 or 3'.

We agree with reviewer #1 and split every sheet of Supplementary Table 2, 3 and 6 into separate Supplementary Tables.

Reviewer #3 (Remarks to the Author):

In the abstract the authors write that RV144 is the vaccine but in the first line of the intro they write that RV144 was a trial. This is confusing, please be consistent (was the trial called the same as the vaccine or what is happening here).

We modified the abstract to clearly distinguish between the RV144 trial and the vaccine ALVAC-HIV + AIDSVAX B/E vaccine (line 27).

In the discussion the authors mentioned the ALVAC vector again which was only mentioned in the introduction. This seems a little random and might deserve a little bit more context.

We agree with reviewer #3 and added text in the discussion reminding readers that ALVAC is the viral vector used to prime RV144 vaccinees (lines 406-407).

Also the authors mentioned that in HIV+ participants the NFkB and mTORC pathway were associated with infection. Their wording that these pathways are required for HIV infection and hence these participants did not benefit from the vaccination is not coherent. I would suggest to reword this slightly to make sure the reader does not implicate causation.

We agree with reviewer #3 and modified that sentence to clarify that some of the genes downstream of NFkB and mTORC1 are host genes important for HIV-1 life cycle and that those genes do not cause HIV-1 infection but are only associated with an increased risk of acquisition (lines 413-414).

There are several supplementary multipage PDFs which I cannot decipher. The file names are inconclusive, and it's a mix of tables and figures. Some tables are completely useless as they cannot be read, others seem more readable however I can't tell what they refer to. Please fix this.

We agree with reviewer #3 and split large Supplementary Tables into separate smaller Supplementary Tables that are more easily readable.

Finally, while I am happy that the authors addressed all comments, I lack structure in doing so. The authors have obviously forgotten to indicate the changed text passages in the rebuttal letter and since the original page and line numbers have certainly changed due to additions to the text, this was hard work. Please make this easier in the future as this delays the review significantly!

We appreciate the hard work of the reviewers and added the line numbers in our rebuttal documents to make their task easier.